# EFFICIENT REWARD POISONING ATTACKS ON ONLINE DEEP REINFORCEMENT LEARNING

## ABSTRACT

We study reward poisoning attacks on online deep reinforcement learning (DRL), where the attacker is oblivious to the learning algorithm used by the agent and does not necessarily have full knowledge of the environment. We demonstrate the intrinsic vulnerability of state-of-the-art DRL algorithms by designing a general, black-box reward poisoning framework called adversarial MDP attacks. We instantiate our framework to construct several new attacks which only corrupt the rewards for a small fraction of the total training timesteps and make the agent learn a low-performing policy. Our key insight is that state-of-the-art DRL algorithms strategically explore the environment to find a high-performing policy. Our attacks leverage this insight to construct a corrupted environment where (a) the agent learns a high-performing policy that has low performance in the original environment and (b) the corrupted environment is similar to the original one so that the attacker's budget is reduced. We provide a theoretical analysis of the efficiency of our attack and perform an extensive evaluation. Our results show that our attacks efficiently poison agents learning with a variety of state-of-the-art DRL algorithms, such as DQN, PPO, SAC, etc., under several popular classical control and MuJoCo environments.

## 1 INTRODUCTION

In several important applications such as robot control (Christiano et al., 2017) and recommendation systems (Afsar et al., 2021; Zheng et al., 2018), state-of-the-art online deep reinforcement learning (DRL) algorithms rely on human feedbacks in terms of rewards, for learning high-performing policies. This dependency raises the threat of reward-based data poisoning attacks during training: a user can deliberately provide malicious rewards to make the DRL agent learn low-performing policies. Data poisoning has already been identified as the most critical security concern when employing learned models in industry (Kumar et al., 2020). Thus, it is essential to study whether state-of-the-art DRL algorithms are vulnerable to reward poisoning attacks to discover potential security vulnerabilities and motivate the development of more robust training algorithms.

**Challenges in poisoning DRL agents.** To uncover practical vulnerabilities, it is critical that the attack does not rely on unrealistic assumptions about the attacker's capabilities. Therefore for ensuring a practically feasible attack, we require that: (i) the attacker has no knowledge of the exact DRL algorithm used by the agent as well as the parameters of the neural network used for training. Further, it should be applicable to different kinds of learning algorithms (e.g., policy optimization, Q learning) (ii) the attacker does not have detailed knowledge about the agent's environment, and (iii) to ensure stealthy, the amount of reward corruption applied by the attacker is limited (see Section 3). As we show in Appendix G, these restrictions make finding an efficient attack very challenging.

**This work: efficient poisoning attacks on DRL.** To the best of our knowledge, no prior work studies the vulnerability of the DRL algorithms to reward poisoning attacks under the practical restrictions mentioned above. To overcome the challenges in designing efficient attacks and demonstrate the vulnerability of the state-of-the-art DRL algorithms, we make the following contributions:

1. We propose a general, efficient, and parametric reward poisoning framework for DRL algorithms, which we call adversarial MDP attack, and instantiate it to generate several attack methods that are applicable to any kind of learning algorithms and computationally efficient. To the best of our knowledge, our attack is the first one that considers the following four key elements in the

threat model at the same time: 1. Training time attack, 2. Deep RL, 3. Reward poisoning attack, 4. Complete black box attack (no knowledge or assumption about the learning algorithm and the environment). A detailed explanation for each key point is provided in Appendix A.

2. We provide a theoretical analysis of our attack methods based on certain assumptions on the efficiency of the DRL algorithms which yields several insightful implications.

3. We provide an extensive evaluation of our attack methods for poisoning the training with several state-of-the-art DRL algorithms, such as DQN, PPO, SAC, etc., in the classical control and MuJoCo environments, commonly used for developing and testing DRL algorithms. Our results show that our attack methods significantly reduce the performance of the policy learned by the agent in the majority of the cases and are considerably more efficient than baseline attacks (e.g., VA2C-P (Sun et al., 2020), reward-flipping (Zhang et al., 2021b)). We further validate the implications of our theoretical analysis by observing the corresponding phenomena in experiments.

## 2    RELATED WORK

**Testing time attack on RL.** Testing time attack (evasion attack) in deep RL setting is popular in literature (Huang et al., 2017; Kos & Song, 2017; Lin et al., 2017). For an already trained policy, testing time attacks find adversarial examples where the learned policy has undesired behavior. In contrast, our training time attack corrupts reward to make the agent learn low-performing policies.

**Data poisoning attack on bandit and tabular RL settings.** Jun et al. (2018); Liu & Shroff (2019); Xu et al. (2021b) study data poisoning attack against bandit algorithms. Ma et al. (2019) studies the attack in the offline tabular RL setting. Rakhsha et al. (2020); Zhang et al. (2020b) study the online tabular RL setting relying on full or partial knowledge of the environment and the learning algorithm. Liu & Lai (2021); Xu et al. (2021a) discuss the attack that can work with no knowledge or weak assumptions on the learning algorithm or the environment. Both tabular and bandit settings are simpler than the deep RL setting considered in our work.

**Observation perturbation attack and defense.** There is a line of work studying observation perturbation attacks during training time (Behzadan & Munir, 2017a;b; Inkawhich et al., 2019) and the corresponding defense (Zhang et al., 2021a; 2020a). The threat model here does not change the actual state or reward of the environment, but instead, it changes the learner's observation of the environment by generating adversarial examples. In contrast, for the poisoning attack as considered in our work, the actual reward or state of the environment is changed by the attacker. The observation perturbation attack assumes access to perturb the sensor of the agent that is used to observe the environment. Therefore, it is not practical when the attacker does not have access to the agent's sensor, or the agent does not rely on sensors for interacting with the environment.

**Data poisoning attack on DRL.** The work of Sun et al. (2020) is the only other work that considers reward poisoning attack on DRL and therefore is the closest to ours. There are three main limitations of their attack compared to ours (a) the attack requires the knowledge of the learning algorithm (the update rule for learned policies) used by the agent, which is not the complete black box setting, (b) the attack only works for on-policy learning algorithms, and (c) the attacker in their setting makes the decision about attacking after receiving a whole training batch. This makes the attack infeasible when the agent updates the observation at each time step, as in this case it is impossible for the attacker to apply corruption to previous observations in a training batch. We experimentally compare against them by adapting our general attacks to their restricted setting. Our results in Appendix I show that our attack requires much less computational resources and achieves better attack results.

**Robust learning algorithms against data poisoning attack.** Robust learning algorithms can guarantee efficient learning under the data poisoning attack. There have been studies on robustness in the bandit (Lykouris et al., 2018; Gupta et al., 2019), and tabular MDP settings (Chen et al., 2021; Wu et al., 2021; Lykouris et al., 2021), but these results are not applicable in the more complex DRL setting. For the DRL setting, Zhang et al. (2021b) proposes a learning algorithm guaranteed to be robust in a simplified DRL setting under strong assumptions on the environment (e.g., linear Q function and finite action space). The algorithm is further empirically tested in actual DRL settings, but the attack method used for testing robustness, which we call reward flipping attack, is not very efficient and malicious as we show in Appendix H. Testing against weak attack methods can provide a false sense of security. Our work provides attack methods that are more suitable for empirically measuring the robustness of learning algorithms.

## 3 BACKGROUND

**Reinforcement learning.** We consider a standard RL setting where an agent is trained by interacting with an environment. The interaction involves the agent observing a state representation of the environment, taking an action, and receiving a reward. Formally, an environment is represented by a Markov decision process (MDP), $\mathcal{M} = \{\mathcal{S}, \mathcal{A}, \mathcal{P}, \mathcal{R}, \mu\}$, where $\mathcal{S}$ is the state space, $\mathcal{A}$ is the action space, $\mathcal{P}$ is the state transition function, $\mathcal{R}$ is the reward function, and $\mu$ is the distribution of the initial states. The training process consists of multiple episodes where each episode is initialized with a state sampled from $\mu$, and the agent interacts with the environment in each episode until it terminates. A policy $\pi : \mathcal{S} \rightarrow D(\mathcal{A})$ is a mapping from the state space to the space of probability distribution $D(\mathcal{A})$ over the action space. If a policy $\pi$ is deterministic, we use $\pi(s)$ to represent the action it suggests for state $s$. A value function $V_{\mathcal{M}}^{\pi}(s)$ is the expected reward an agent obtains by following the policy $\pi$ starting at state $s$ in the environment $\mathcal{M}$. We denote $\mathcal{V}_{\mathcal{M}}^{\pi} := \mathbb{E}_{s_0 \sim \mu} V_{\mathcal{M}}^{\pi}(s_0)$ as the policy value for a policy $\pi$ in $\mathcal{M}$, which measures the performance of $\pi$. The goal of the RL agent is to find the optimal policy with the highest policy value $\pi^* = \arg\max_{\pi} \mathcal{V}_{\mathcal{M}}^{\pi}$. For ease of analysis, the state distribution is defined as $\mu^{\pi}(s) = \mathbb{E}_{\pi, s_0 \sim \mu}[\sum_t \mathbb{1}[s^t = s]]$, representing how often a state is visited under policy $\pi$ in an episode.

**Reward poisoning attack on deep RL.** In this work we consider a standard data poisoning attack setting (Jun et al., 2018; Rakhsha et al., 2020) where a malicious adversary tries to manipulate the agent by poisoning the reward received by the agent from the environment during training. The attacker observes the current state, action, and reward tuple $(s^t, a^t.r^t)$ generated during training at each timestep $t$ and injects a corruption $\Delta^t$ on the true reward $r^t$. As a result, the environment returns the agent with the corrupted observation $(s^t, a^t, s^{t+1}, r^t + \Delta^t)$ where $s^{t+1}$ is the next state. Next, we describe the restrictions on the attacker's capabilities as mentioned in the introduction:

1. **Limited budget.** The attacker can only corrupt a small number of timesteps $C$, i.e., $\sum_{t=0}^{T} \mathbb{1}\{\Delta^t \neq 0\} \leq C$ and $C \ll T$ where $T$ is the total number of training steps.

2. **Limited per-step corruption.** The corruption at each timestep is limited by $|\Delta^t| \leq B, \forall t \in [T]$.

3. **Limited per-episode corruption:** The total corruption across an episode is limited by $\sum_{t \in t^e} |\Delta^t| \leq E$ where $t^e$ is the set of all timesteps in an episode $e$.

4. **Oblivious of the DRL algorithm.** The attacker has no knowledge of the training algorithm or any parameters in the network used by the agent while training.

5. **Oblivious of the environment.** The attacker has no knowledge about the MDP $\mathcal{M}$ except for the number of dimensions and range of each dimension in the state and action space $\mathcal{S}, \mathcal{A}$. Our attacks do not need knowledge of $\mathcal{M}$ but can benefit from having access to a good performing policy in $\mathcal{M}$ which could be trained by itself, another agent, or publicly available. We consider these additional cases to study the impact of increasing attacker resources on its efficiency.

Let $\pi_0$ be the best learned policy when the DRL training finishes, the goal of the attacker is to corrupt training such that the performance of the learned policy $\pi_0$ in the environment $\mathcal{M}$: $\mathcal{V}_{\mathcal{M}}^{\pi_0}$ is low. Note that $\mathcal{V}_{\mathcal{M}}^{\pi}$ is an intrinsic property of a given policy $\pi$ in $\mathcal{M}$ and is independent of the learning algorithm. Reward poisoning does not change $\mathcal{V}_{\mathcal{M}}^{\pi}$ but instead makes the agent learn a policy $\pi$ with lower $\mathcal{V}_{\mathcal{M}}^{\pi}$. For clarity, we summarize the goal, knowledge, and constraints for our attacker in Appendix B. We consider multiple constraints on the attacker to make the setting more realistic defined above. The full constraints are considered in experiments in Section 6. For the purpose of theoretical analysis, we simplify the problem by dropping the constraint on per-episode corruption $E$ in Section 4 and Section 5.

## 4 FORMULATING REWARD POISONING ATTACK

A reward poisoning attack algorithm can be represented by its attack strategy $A^t$ at each timestep during training. An attack strategy $A^t$ depends on the full observation before that attack, that is, all the states $s^{1:t}$, actions $a^{1:t}$, rewards $r^{1:t-1}$. The output of $A^t$, i.e, the corruption on reward, satisfies $\Delta^t = A^t(s^{1:t}, a^{1:t}, r^{1:t-1})$. A practical attack should be constructed in a computationally efficient manner and should work in a practical setting where it is oblivious to the environment and learning algorithm. In Appendix G we show that searching for the optimal or near-optimal attack is computationally hard regardless of the attack constraints in the DRL setting and requires full knowledge of the learning algorithm and environment. Since it is hard to construct both optimal and

practical attacks, we focus on finding feasible attacks that are not necessarily the optimal ones but still make a learning algorithm learn a policy with low policy value with a limited budget. Formally, the attacker's objective, i.e., finding a feasible attack, can be stated using the following constraints:

$$\text{find } \Delta^{t=1,\dots,T} \text{ s.t. } \mathcal{V}_{\mathcal{M}}^{\pi_0} \leq V; \sum_{t=1}^{T} \mathbb{1}[\Delta^t \neq 0] \leq C; |\Delta^t| \leq B, \forall t \in [T]. \tag{1}$$

To solve equation 1, one way is to fix the values of $V$, $B$, and $C$ and solve for an attack $\Delta^t$. However, it is possible that no solution is feasible for certain $V$, $B$, and $C$. Since each feasibility check can be expensive, we look for a more efficient and convenient way to solve equation 1. We fix the attack and estimate the corresponding $V$, $B$, and $C$ to satisfy equation 1. An attack algorithm is efficient if it can satisfy equation 1 with low values for $B, C, V$ with $V < \mathcal{V}_{\mathcal{M}}^{\pi^*}$ otherwise the attack is trivial as $\mathcal{V}_{\mathcal{M}}^{\pi_0} \leq \mathcal{V}_{\mathcal{M}}^{\pi^*}$ holds without any attack. As confirmed by our theoretical analysis in Section 5 and experiments in Section 6, finding an efficient attack based on equation 1 is non-trivial.

**Adversarial MDP attack.** To find attack algorithms for solving equation 1, we introduce a general parametric attack framework called "adversarial MDP attack" for poisoning the training of deep RL agents. The high-level idea behind our attack is to construct a fixed adversarial environment to train the agent. This idea has been applied in designing attacks in the simpler bandit (Liu & Shroff, 2019) and tabular setting Rakhsha et al. (2020) where they formulate the problem of finding the best adversarial environment for their attack goal as an optimization problem and solve it directly. Solving such an optimization problem is computationally infeasible in the deep RL setting due to the complexity of both the environment and the learning algorithm. Therefore we design new efficient algorithms that are suited to the deep RL setting and our attack scenario. In our attack, the attacker constructs an adversarial MDP $\widehat{\mathcal{M}} = \{\mathcal{S}, \mathcal{A}, \mathcal{P}, \widehat{\mathcal{R}}, \mu\}$ for the agent to train on by injecting the corrupted reward $\widehat{\mathcal{R}}$ to the environment during training. More specifically, for an adversarial MDP attack with $\widehat{\mathcal{M}}$, its attack strategy at round $t$ only depends on the agent's current state and action:

$$\Delta^t = A^t(s^t, a^t) = \hat{R}(s^t, a^t) - R(s^t, a^t) \tag{2}$$

Next, we compute bounds on $V$, $B$, and $C$ such that $A^{1:T}$ constructed using equation 2 from a given $\widehat{\mathcal{M}}$ in our framework is a feasible solution to equation 1. The lower these values, the more efficient is the adversarial MDP $\widehat{\mathcal{M}}$. We make two simplifying assumptions: (i) the learning algorithm can always find and report the optimal policy from a fixed environment, i.e.,$\pi_0 = \pi^*$ always holds. We note that our experimental results show that our attack succeeds with low values of V, $B$ and $C$ on state-of-the-art deep RL algorithms that do not always learn an optimal policy, and (ii) the learning algorithm explores strategically, that is, instead of uniformly exploring all state-action pairs, the algorithm does not waste many rounds to explore the state action pairs that have little value. This assumption is satisfied by RL algorithms (Dong et al., 2019; Jin et al., 2018; Agarwal et al., 2019) and also validated in our experiments.

**Lower bounds on V, B, and C for a given $\widehat{\mathcal{M}}$.** The bound on $V$ relates to the optimal policy $\hat{\pi}^* := \arg\max_\pi \mathcal{V}_{\widehat{\mathcal{M}}}^{\pi}$ that the algorithm will learn under $\widehat{\mathcal{M}}$ based on our first assumption, i.e., $\pi_0 = \hat{\pi}^*$. We can directly bound $V$ by the policy value of $\hat{\pi}^*$ under $\mathcal{M}$: $V \geq \mathcal{V}_{\mathcal{M}}^{\hat{\pi}^*}$. It is straightforward to bound $B$ as $B \geq ||\widehat{\mathcal{R}} - \mathcal{R}||_\infty$. For bounding $C$, our attack applies corruption whenever the learning algorithm chooses an action $a^t$ at a state $s^t$ such that the reward function at this state action pair are different for the real and adversarial MDP, i.e, $\hat{\mathcal{R}}(s^t, a^t) \neq \mathcal{R}(s^t, a^t)$. Then $C$ can be bound as $C \geq \sum_{t=1}^{T} \mathbb{1}[\hat{\mathcal{R}}(s^t, a^t) \neq \mathcal{R}(s^t, a^t)]$. Under our second assumption on strategic exploration of the learning algorithm, the bound on $C$ will be low if $\widehat{\mathcal{M}}$ satisfies $\widehat{\mathcal{R}}(s, \hat{\pi}^*(s)) = \mathcal{R}(s, \hat{\pi}^*(s))$ for all states $s$, as for most of the time $a^t = \hat{\pi}^*(s)$, resulting in $\mathbb{1}[\hat{\mathcal{R}}(s^t, a^t) \neq \mathcal{R}(s^t, a^t)] = 0$.

**Effect of given $V, B$ and $C$ on $\widehat{\mathcal{M}}$.** Raising the value of $B$ or $C$ increases the number of adversarial MDP's that satisfy equation 1. To achieve the same value of $V$, a larger value of $B$ can reduce the requirement on $C$ to ensure existance of feasible solutions, and vice versa. Next, we will instantiate our framework to construct specific $\widehat{\mathcal{M}}$ that result in efficient attacks which significantly reduce the performance of the learned policy under a low budget.

## 5 POISONING ATTACK METHODS

In this section, we design new reward poisoning attacks for deep RL by instantiating our adversarial MDP attack framework. Each instantiation provides a parameterized way to construct an adversarial MDP $\widehat{\mathcal{M}}$ via a parameter $\Delta$, corresponding to the amount of reward corruption for poisoning applied by the attacker at a timestep. The definition of $\Delta$ yields $|\Delta| = ||\hat{\mathcal{R}} - \mathcal{R}||_\infty$, resulting in a lower bound on requirement $B \geq |\Delta|$. For a given value of V, let $\Delta(V)$ be the minimum absolute value of the parameter $\Delta$ for the instantiation to satisfy the requirement on $V$ in equation 1. We note that $\Delta(V)$ corresponds to the minimum requirement on $B$ for the attack to satisfy $V$ in equation 1. We provide symbolic expressions for $\Delta(V)$ corresponding to each instantiation and then use it to construct an upper bound on $\Delta(V)$, which is easier to reason about than exact expressions. We define $G_V^{\mathcal{M}}$ and $B_V^{\mathcal{M}}$ with respect to value $V$ to be the set of all policies with policy value $> V$ and $\leq V$ respectively under $\mathcal{M}$. The upper bound on $\Delta(V)$ will be constructed using $G_V^{\mathcal{M}}$ and $B_V^{\mathcal{M}}$. We will empirically examine the requirement on $C$ for each instantiation in Section 6. An attack is efficient if to satisfy a certain value of $V$ in equation 1, it requires low value of $C$ and $B = \Delta(V)$. All the attack methods we are going to propose do not require any knowledge about both the learning algorithm and the environment, though one method gain benefits from full or partial knowledge of the environment as we find in practice. For simplicity, we start with attacks on environments with discrete action space and then show how to transfer the attack and the corresponding analysis to continuous action space. Note that our analysis in this section is based on the two assumptions we made in Section 4 that the learning algorithms can always learn the optimal policy and explore strategically. All the proofs for the theorems and lemmas can be found in the Appendix E.

**Uniformly random time(UR) attack.** We first introduce a trivial instantiation of adversarial MDP attack framework as a baseline, which we call the UR attack. Here, the attacker randomly decides whether to corrupt the reward at a timestep with a fixed probability $p$ and amount $\Delta$ regardless of the current state and action. Formally, the attack strategy of the UR attacker at time $t$ is: $A^t(s^t, a^t) = \Delta$ with probability $p$, otherwise $A^t(s^t, a^t) = 0$. Let $V_{\max} = \max_\pi \mathcal{V}_{\mathcal{M}}^\pi$ and $V_{\min} = \min_\pi \mathcal{V}_{\mathcal{M}}^\pi$ be the maximum and minimum policy value computed over all policies under $\mathcal{M}$. We provide the following bounds on $\Delta(V)$ for the UR attack:

**Theorem 5.1.** *For the UR attack parameterized with probability $p$, the exact expression and an upper bound on $\Delta(V)$ are:*

$$\Delta(V) = \min_{\pi_1 \in B_V^{\mathcal{M}}} \max_{\pi_2 \in G_V^{\mathcal{M}}} \frac{\mathcal{V}_{\mathcal{M}}^{\pi_2} - \mathcal{V}_{\mathcal{M}}^{\pi_1}}{p \cdot |L^{\pi_1} - L^{\pi_2}|} \leq \frac{V_{\max} - V_{\min}}{p \cdot \min_{\pi_1 \in G_V^{\mathcal{M}}} \max_{\pi_2 \in B_V^{\mathcal{M}}} |L^{\pi_1} - L^{\pi_2}|}$$

*where $L^\pi := \sum_s \mu^\pi(s)$ is the expected length of an episode for an agent following the policy $\pi$. Moreover, the expression only holds for $\Delta$ with the same sign as $L^{\pi_1} - L^{\pi_2}$ for the $\pi_1$, $\pi_2$ that realize the min-max condition.*

Theorem 5.1 shows that the value of $\Delta(V)$ depends on the difference in episode length from policies in $G_V^{\mathcal{M}}$ and $B_V^{\mathcal{M}}$. A low value of $p$ also makes high value of $\Delta(V)$. In addition, the sign of $\Delta$ needs to be chosen correctly to make $\mathcal{V}_{\mathcal{M}}^{\hat{\pi}^*} < \mathcal{V}_{\mathcal{M}}^{\pi^*}$, otherwise it can make the optimal policies look even better under $\widehat{\mathcal{M}}$, which yields the following implications:

> **Implication 1.** Compared to the UR attack in the right direction, the UR attack in the wrong direction requires a higher value of $\Delta$ to make the learning agent learn the policy with the same performance (expected reward per episode), or even worse: the attack in the wrong direction can never prevent the learning agent from finding the optimal policy.

Implication 1 suggests that for the UR attack, it is important to find the right direction for corruption.

**Learned policy evasion (LPE) attack.** The high-level idea behind the LPE attack is to make all policies of good performance appear bad to the learning agent. Intuitively, policies of good performance should share similar behavior as there is usually a certain general strategy to behave well in the environment. Therefore if the attacker can make the actions corresponding to such behavior look bad, then all the good policies will appear bad to the agent. Formally, the LPE attack is characterized by the policy $\pi^\dagger$ available to the attacker, and it penalizes the learning algorithm whenever it chooses an action that corresponds to $\pi^\dagger$. Correspondingly, the attack strategy is: $A^t(s^t, a^t) = \Delta \cdot \mathbb{1}\{a^t = \pi^\dagger(s^t)\}$, where $\Delta < 0$ is a fixed value. $\pi^\dagger$ is learned offline by the

attacker before the agent starts learning. We will show different ways the attacker can generate $\pi^\dagger$ in Section 6. Next, we analyze the efficiency of the attack. To help our analysis, we introduce the following definition to measure the similarity $D(\pi_1, \pi_2)$ between two policies $\pi_1$ and $\pi_2$:

**Definition 5.2.** (Similarity between policies) The similarity of a policy $\pi_1$ to a policy $\pi_2$ is: $D(\pi_1, \pi_2) = \sum_{s \in \mathcal{S}} \mu^{\pi_1}(s) \mathbb{1}[\pi_1(s) = \pi_2(s)]$.

The similarity of $\pi_1$ to $\pi_2$ increases with the frequency with which $\pi_2$ takes the same action as $\pi_1$ in the same state $s$. Note that $D(\pi_1, \pi_2) \neq D(\pi_2, \pi_1)$, and $D(\pi_1, \pi_2) \leq L^{\pi_1}$.

**Theorem 5.3.** *For LPE attack with $\pi^\dagger$, the expression and an upper bound on $\Delta(V)$ are:*

$$\Delta(V) = \min_{\pi_1 \in B_V^{\mathcal{M}}} \max_{\pi_2 \in G_V^{\mathcal{M}}} \frac{\mathcal{V}_{\mathcal{M}}^{\pi_2} - \mathcal{V}_{\mathcal{M}}^{\pi_1}}{D(\pi_2, \pi^\dagger) - D(\pi_1, \pi^\dagger)} \leq \frac{V_{\max} - V_{\min}}{\min_{\pi \in G_V^{\mathcal{M}}} D(\pi, \pi^\dagger) - \min_{\pi \in B_V^{\mathcal{M}}} D(\pi, \pi^\dagger)}.$$

In Appendix E we will show that $\min_{\pi \in B_V^{\mathcal{M}}} D(\pi, \pi^\dagger)$ is likely to be 0 in general cases. Theorem 5.3 shows that the requirement on $\Delta$ for the LPE attack is inversely proportional to the minimum similarity between $\pi^\dagger$ and a policy from $G_V^{\mathcal{M}}$. In practice, we observe that in most cases there are usually certain behaviors shared in common by the non-trivial policies that have better performance than random ones. This yields the following implications:

> **Implication 2.** With the same value of $\Delta$, the LPE attack can make the agent learn policies of worse performance with high performing $\pi^\dagger$ compared to the LPE attack with a random policy.

The LPE attack can generate a random policy as $\pi^\dagger$. For experiments in Section 6, to simplify implementation, we generate the random policy through random initialization of a learning algorithm different from the one used by the agent. When the attacker has access to a high-performing policy, it can be used as $\pi^\dagger$ which may provide better attack performance. Implication 2 suggests that the LPE attack should use a high-performing policy to attack if possible. To estimate the requirement on $C$, we give the following lemma:

**Lemma 5.4.** *For the LPE attack, $\widehat{\mathcal{R}}(s, \hat{\pi}^*(s)) = \mathcal{R}(s, \hat{\pi}^*(s)), \forall s \in \mathcal{S} \iff D(\pi^\dagger, \hat{\pi}^*) = 0$. Given sufficient $\Delta$, the optimal policy under $\widehat{\mathcal{M}}$ given by the LPE attack with $\pi^\dagger$ satisfies $D(\pi^\dagger, \hat{\pi}^*) = 0$.*

Recall that in Section 4, we estimate that the requirement on $C$ will be low if the adversarial and real reward are the same at all $(s, \hat{\pi}^*(s))$, i.e., $\widehat{\mathcal{R}}(s, \hat{\pi}^*(s)) = \mathcal{R}(s, \hat{\pi}^*(s))$. Lemma 5.4 suggests that this condition will hold for the LPE attack given sufficient $\Delta$, as for the LPE attack $D(\pi^\dagger, \hat{\pi}^*) = 0 \iff \widehat{\mathcal{R}}(s, \hat{\pi}^*(s)) = \mathcal{R}(s, \hat{\pi}^*(s))$. This yields

> **Implication 3:** Under the LPE attack with sufficient $|\Delta|$, the learning agent will gradually converge to $\hat{\pi}^*$ under $\widehat{\mathcal{M}}$ eventually and few corruptions will be applied afterward, resulting in a decrease in attack frequency as the training goes on.

**Random policy inducing (RPI) attack.** The intuitive idea behind the RPI attack is to make the agent believes that a random policy is an optimal one. To achieve this, the attacker can make all the actions that are different from the ones given by the random policy look bad. The RPI attack is characterized by a randomly generated policy $\pi^\dagger$, and it penalizes the learning algorithm whenever it doesn't follow the action that corresponds to $\pi^\dagger$. Formally, the attack strategy of the RPI attack with a policy $\pi^\dagger$ is: $A^t(s^t, a^t) = \Delta \cdot \mathbb{1}\{a^t \neq \pi^\dagger(s^t)\}$, where $\Delta < 0$ is a fixed value. We have the following expression and upper bound on $\Delta(V)$:

**Theorem 5.5.** *For RPI attack with policy $\pi^\dagger$, the expression and an upper bound on $\Delta(V)$ are:*

$$\Delta(V) = \min_{\pi_1 \in B_V} \max_{\pi_2 \in G_V} \frac{\mathcal{V}_{\mathcal{M}}^{\pi_2} - \mathcal{V}_{\mathcal{M}}^{\pi_1}}{(L^{\pi_2} - L^{\pi_1}) - (D(\pi_2, \pi^\dagger) - D(\pi_1, \pi^\dagger))} \leq \frac{V_{\max} - V_{\min}}{\min_{\pi \in G_V^{\mathcal{M}}}(L^\pi - D(\pi, \pi^\dagger))}$$

Theorem 5.5 suggests that the requirement on $B$ will be low if all policies in $G_V^{\mathcal{M}}$ have low similarity to $\pi^\dagger$. With the same observation we give about the similarity between policies better than the random ones, it suggests that $\Delta(V)$ will be less when $\pi^\dagger$ is a random policy. Then theorem 5.5 gives

**Implication 4:** With the same value of $\Delta$, the RPI attack with random $\pi^\dagger$ can make the learning algorithm learn policies of worse performance than the RPI attack with high performing $\pi^\dagger$.

Implication 4 suggests that the RPI attack should always use a random policy for the attack. To estimate the requirements on $C$, we give the following lemma:

**Lemma 5.6.** *For the RPI attack, $\widehat{\mathcal{R}}(s, \hat{\pi}^*(s)) = \mathcal{R}(s, \hat{\pi}^*(s)) \iff \hat{\pi}^* = \pi^\dagger$. Given sufficient $\Delta$, the optimal policy under $\widehat{\mathcal{M}}$ constructed by the RPI attack with $\pi^\dagger$ is $\hat{\pi}^* = \pi^\dagger$.*

Lemma 5.6 suggests that with a sufficient value of $|\Delta|$, $\widehat{\mathcal{R}}(s, \hat{\pi}^*(s)) = \mathcal{R}(s, \hat{\pi}^*(s))$ will hold, resulting in low requirement on $C$. This also implies that

**Implication 5:** For the RPI attack with sufficient $|\Delta|$, the learning agent will gradually converge to $\pi^\dagger$, resulting in a decrease in the frequency of corruption as the training goes on.

Our two main attack methods LPE and RPI proposed so far work with negative corruption on reward and avoid attacking the action corresponding to $\hat{\pi}^*$ in every state. The methods will work well under our assumptions on learning algorithms. An alternative is to attack with positive corruption on the reward. Under the framework of adversarial MDP attack, such an attack can result in less requirements on $C$ when the learning agent does the opposite to our second assumption, i.e., exploring the sub-optimal actions more often. To compare the difference in performance between these types of attack, we propose a variant of RPI attack called random policy promoting (RPP) attack.

**Random policy promoting (RPP) attack.** The RPP attack shares the same intuition about highlighting a random policy, but instead, it positively rewards the actions corresponding to the random policy. Formally, the attack strategy of RPP attack with a policy $\pi^\dagger$ is $A^t(s^t, a^t) = \Delta \cdot \mathbb{1}\{a^t = \pi^\dagger(s^t)\}$, where $\Delta > 0$ is a fixed value. The expression and an upper bound on $\Delta(V)$ are:

**Theorem 5.7.** *Under RPP attack with policy $\pi^\dagger$, the expression and an upper bound for $\Delta(V)$ are:*

$$\Delta(V) = \min_{\pi_1 \in B_V} \max_{\pi_2 \in G_V} \frac{\mathcal{V}_{\mathcal{M}}^{\pi_2} - \mathcal{V}_{\mathcal{M}}^{\pi_1}}{D(\pi_2, \pi^\dagger) - D(\pi_1, \pi^\dagger)} \leq \frac{V_{\max} - V_{\min}}{\max\{L^{\pi^\dagger} - \max_{\pi \in G_V^{\mathcal{M}}} D(\pi, \pi^\dagger), 0\}}$$

We note that compared to the RPI attack, the RPP attack requires more $B$ if $L^{\pi^\dagger} < \min_{\pi \in G_V^{\mathcal{M}}} L^\pi$, as it results in $L^{\pi^\dagger} - \max_{\pi \in G_V^{\mathcal{M}}} D(\pi, \pi^\dagger) < \min_{\pi \in G_V^{\mathcal{M}}} L^\pi - \max_{\pi \in G_V^{\mathcal{M}}} D(\pi, \pi^\dagger) \leq \min_{\pi \in G_V^{\mathcal{M}}} (L^\pi - D(\pi, \pi^\dagger))$. Further if $L^{\pi^\dagger} - \max_{\pi \in G_V^{\mathcal{M}}} D(\pi, \pi^\dagger)$ becomes less than 0, then equation 1 can never be satisfied with any value of $B$. This happens when a policy in $G_V^{\mathcal{M}}$ benefits more from the positive corruption than policies from $B_V^{\mathcal{M}}$. This implies that

**Implication 6:** For environments where policies of high values are associated with long episodes, the RPI attack can make the learning agent learn worse policy than the RPP attack, and the RPP attack may even be ineffective regardless of $\Delta$.

As mentioned above, the requirement on $C$ for the RPP and RPI attack depends on how much the learning algorithm deviates from our second assumption in Section 4. From experiments in Section 6, under the same constraints on the attack, we observe that for most learning algorithms, the RPI attack usually achieves better attack results. Our results suggest that although the RPP attack can be more efficient for certain learning algorithms and environments, the RPI attack is in general more reliable and efficient than the RPP attack.

Finally, note that the upper bound we provide for $\Delta(V)$ of all attacks here satisfies $\Delta(V) \geq (V_{\max} - V_{\min})/L_{\max}$, where $L_{\max} := \max_\pi L^\pi$ is the maximum episode length. Therefore for experiments in Section 6, we always let $|\Delta| > (V_{\max} - V_{\min})/L_{\max}$.

**Extension to environments with continuous action space.** To extend the above attacks from discrete to continuous action space, we adaptively discretize the continuous action space with respect to the action from the learning agent. Formally, we consider two actions the same if their distance in the action space is less than a given threshold $r$. Then the aforementioned attack methods can decide whether to apply corruption based on whether two actions are considered the same given $r$. For example, the attack strategy for the LPE attack with $\pi^\dagger$ in continuous action space is $A^t = \Delta \cdot \mathbb{1}[||a^t - \pi^\dagger(s^t)||_2 \leq r]$. Accordingly, we define the similarity between policies for continuous action space parameterized by $r$:

**Definition 5.8.** (Similarity between policies under distance $r$) The similarity of a policy $\pi_1$ to a policy $\pi_2$ in continuous action space parameterized by $r$ is defined as $D_c(\pi_1, \pi_2, r) = \sum_{s \in \mathcal{S}} \mu^{\pi_1}(s) \mathbb{1}[\|\pi_1(s) - \pi_2(s)\|_2 \le r]$.

By replacing $D(\pi_1, \pi_2)$ with $D_c(\pi_1, \pi_2, r)$, we can transfer the analysis for the attack from discrete action space to continuous action space. Note that while we measure the distance in L2-norm, any other norm will also work. The extension adds an additional parameter $r$ for the attack methods. In Appendix D we show how $r$ influences the attacks.

## 6 EXPERIMENTS

We evaluate our attack methods from Section 5 for poisoning training with state-of-the-art DRL algorithms in both the discrete and continuous settings. We consider learning in environments typically used for assessing the performance of the DRL algorithms in the literature. As for the implications in Section 5, in Appendix C we experimentally show that they hold when attacking practical DRL algorithms even though they do not necessarily satisfy our assumptions from Section 4.

**Learning algorithms and environments.** We consider 4 common Gym environments (Brockman et al., 2016) in the discrete case: CartPole, LunarLander, MountainCar, and Acrobot, and 4 continuous cases: HalfCheetah, Hopper, Walker2d, and Swimmer. The DRL algorithms in the discrete setting are: dueling deep Q learning (Duel) (Wang et al., 2016) and double dueling deep Q learning (Double) (Van Hasselt et al., 2016) while for the continuous case we choose: deep deterministic policy gradient (DDPG), twin delayed DDPG (TD3), soft actor critic (SAC), and proximal policy optimization (PPO). Overall, the 6 algorithms we consider cover the popular paradigms in model-free learning algorithms: policy gradient, Q-learning, and their combination. The implementation of the algorithms is based on the spinningup project (Achiam, 2018).

**Experimental setup.** The attacks set $|\Delta| = B$, and the signs of $\Delta$ are specified in each attack's strategy. We consider more strict and practical constraints (as described in Section 3) on the attacker than in our theoretical analysis. To work with extra constraints, we modify our adversarial MDP attack framework: if applying corruption as per the attack strategy given by the framework in Section 4 will break the constraints on $E$ or $C$ at a time step, i.e. if at time $t$ in episode $e$, $1 + \sum_{\tau=0}^{t-1} \mathbb{1}[A^\tau > 0] > C$ or $|\Delta| + \sum_{\tau \in t_e} |A^t| > E$, then the attacker applies no corruption at that time step. We choose $T$ to ensure that the learning algorithm can converge within $T$ time steps without the attack. We evaluate the effectiveness of our attacks with different values of $C$ such that the ratio $C/T$ is low. Since the attacker is unaware of the learning algorithm in all of our attacks, for each environment, whenever the attacker needs to learn $\pi^\dagger$ offline, it does so with an algorithm different from the agent's learning algorithm. More specifically, for each environment we select a pair of learning algorithms that are most efficient in learning from the environment (without attack), then while we use one of them for the learning agent to train, the other will be used by the attacker to learn a high performing policy or generate a random policy $\pi^\dagger$. Our criteria yield cases where the learning algorithms in a pair belong to different learning paradigms and have different architectures of neural networks. Our results demonstrate that the efficiency of our attack methods does not depend upon the similarity between the learning algorithms.

To determine $E$ and $B$, we note that $V_{\max} - V_{\min}$ represents the maximum environment-specific net reward an agent can get during an episode, and $\frac{(V_{\max} - V_{\min})}{L_{\max}}$ represents the range of average reward at a time step for an agent. We set $E = \lambda_E \cdot (V_{\max} - V_{\min})$, and $B = \lambda_B \cdot \frac{(V_{\max} - V_{\min})}{L_{\max}}$ where $\lambda_E \le 1, \lambda_B > 1$ are normalization constants to ensure that the values of $E$ and $B$ represent similar attack power across different experiments. We choose $\lambda_E \le 1$ to ensure that the corruption in an episode is $\le$ the maximum net reward a policy can achieve during an episode. We choose $\lambda_B > 1$ that ensures $B > \frac{(V_{\max} - V_{\min})}{L_{\max}}$. This is because the upper bounds on $\Delta(V)$ according to our theorems in Section 5 should be $> \frac{(V_{\max} - V_{\min})}{L_{\max}}$ for all attack algorithms. The exact values of $T$, $B$, and $E$ for different environments are in the Appendix D.

**Main results.** A subset of our main results is shown in figure 1. The full set can be found in Appendix D. The $x$ axis is $C/T$; the $y$ axis is the policy value of the best policy the learning algorithm learned after each epoch across the whole training process. The $y$ value at each data point is averaged over 10 experiments under the same setting. For the same constraint on $C$, we consider

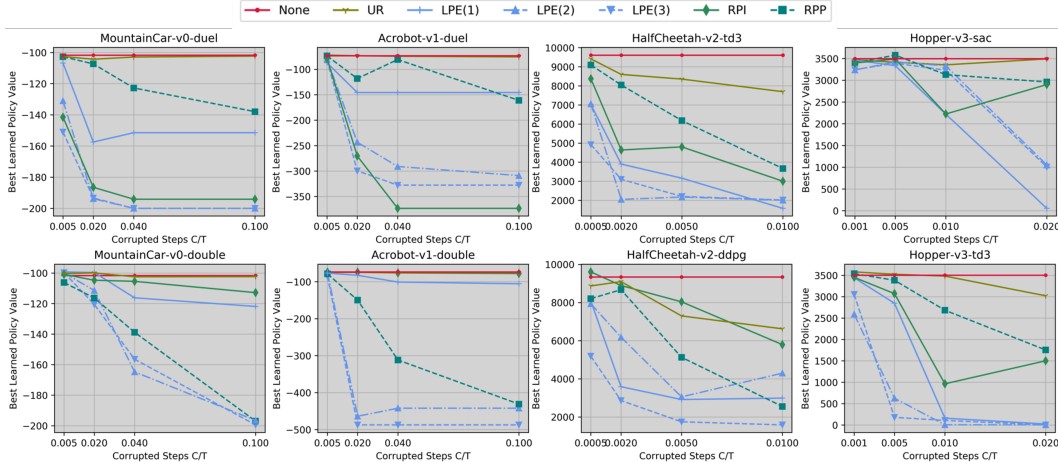

Figure 1: Highest policy value of learned policies by algorithms under no or different attacks.

an attack to be more efficient than another if it has a lower value on the $y$ axis. We consider an attack to be successful if the resulting policy value is lower than the two baselines: the learning agent under no attack and the UR attack. For the UR attack, we set $p = C/T$ so that the corrupted rounds are randomly distributed in the whole training process. We always choose the sign of $\Delta$ that gives the best attack result. In figure 1, the best UR attack has only a small influence on the learning algorithm. We also compare to two other empirical attack methods proposed in previous work (Zhang et al., 2021b; Sun et al., 2020). In Appendix H we explain how the reward flipping attack from Zhang et al. (2021b) works and show that it is no more efficient than the UR attack. In Appendix I we show that the VA2C-P attack in Sun et al. (2020) not only has more limitations as mentioned in Section 2 but also less efficient and is significantly more computationally expensive than our attacks.

For the LPE attack, we consider three variants based on $\pi^\dagger$: (1) the attacker does not have any knowledge about the environment and uses a random policy as $\pi^\dagger$, (2) the attacker trains in the environment for $T$ steps and chooses the best policy as $\pi^\dagger$, and (3) Same as (2) except that the attacker selects a policy as $\pi^\dagger$ which has a policy value that is the closest to the mean of the policy values in the first two cases. This variant is used to check the effect of learning suboptimal $\pi^\dagger$ on the attack performance. We observe in figure 1 that variants (2) and (3) always succeed while (1) fails in 1 case (LunarLander learned by Double). Comparing the three variants, variants (2), and (3) usually achieve better attack results than variant (1). Especially, we notice that LPE (2) achieves the best attack result for training with the Duel algorithm in MountainCar and with the Double algorithm in Acrobot with corruption budget $C/T = 0.04$ and $0.02$ respectively, as the learning algorithms do not learn policies better than random ones with minimal performance. ($-200$ and $-500$ are the minimum rewards from an episode in the two environments respectively)

For the RPI and RPP attacks, they are effective in most cases except that RPI fails in 2 cases (Acrobot learned by Double, and Swimmer learned by PPO which can be found in the full set of results in Appendix D), and RPP fails in 1 case (CartPole learned by Double). Across all cases, the RPI attack usually has better performance than the RPP attack. Especially, we notice that in Acrobot and MountainCar environments, RPI attack achieves better attack results when the learning algorithm is Duel, and the opposite is true when the learning algorithm is Double. This is probably because the Double algorithm does more exploration in suboptimal state action pairs than the Duel algorithm.

## 7 CONCLUSION AND LIMITATIONS

In this work, we studied the security vulnerability of DRL algorithms against training time attacks. We designed a general, parametric framework for reward poisoning attacks and instantiated it to create several efficient attacks. We provide theoretical analysis yielding insightful implications validated by our experiments. Our detailed evaluation confirms the efficiency of our attack methods pointing to the practical vulnerability of popular DRL algorithms. Our attacks have the following limitations: (i) not applicable for other attack goals, e.g, to induce a target policy, (ii) cannot find the optimal attacks, and (iii) do not cover state poisoning attack.

## 8 REPRODUCIBILITY STATEMENT

For the purpose of reproducing the experimental results, we provide the code in the supplementary materials and the necessary instructions in the README file. We provide the experiment details about the setup and hyper-parameters in Section 6 and Appendix D. For checking the correctness of our theoretical analysis, the proof for all theorems and lemmas in Section 5 can be found in Appendix E.

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

## A    KEY ELEMENTS EXPLANATION

We mentioned there are four key elements that our attack methods address at the same time. The four key elements are 1. Training time attack, 2. Deep RL, 3. Reward poisoning, 4. Complete black box. To the best of our knowledge, we are the only paper that covers the above four elements at the same time. Below are the explanations for what each key element means and why it is important to be considered:

1. Training time attack: Testing time attacks target an already learned policy, where the attacker wants to make the learned policy misbehave by crafting examples, such as an adversarial state where the policy suggests a sub-optimal action. The target of training time attacks is a learning algorithm trying to learn a policy, where the attacker wants to make the learning agent learn a policy having undesired behavior by corrupting the training process. In this work we propose efficient training time attacks against DRL so that practical threat against DRL is better understood.

2. Deep RL setting: Most of the works on training time attacks study simpler learning settings like bandit and tabular MDP cases. Considering that DRL is more practical in real-world applications, we study training time attacks in the deep MDP settings.

3. Reward poisoning attack: The threat models for RL include three types of poisoning: state, action, and reward poisoning, and all of them have been studied in different literature. In this work we focus on reward poisoning as it is a more practical threat in applications where the agent collects reward from a human user.

4. Complete black box: In practical cases such as a recommendation system, the learning agent needs to formulate the MDP by itself, which is private to the agent and unknown to the attacker. It is also likely that the attacker does not have any knowledge of the learning algorithm as it is also private information held by the learning agent. To model realistic attacks, we consider attackers with no prior knowledge about both the environment and the learning algorithm.

5. Real time attack: To work in a real-time manner, the attacker should satisfy the two following conditions: 1. It is able to compute the corruption at each time step in a short time. 2. It need to decide and apply corruption at each timestep after the

## B    A SUMMARY OF THE INFORMATION ABOUT THE ATTACKER

For clarity, we summarize all information about the attacker in our setting, including its knowledge, constraint, and goal.

Table 1: The knowledge, constraint, and goal of the attacker

|  | Description |
|---|---|
| Knowledge | 1. Has no information about agent's learning algorithm |
|  | 2. Only aware of the state and action spaces of the environment. |
|  | 3. Can observe the true state, action, and reward at each timestep during the training process. |
| Constraints | 1. corruption can be injected at each step is limited |
|  | 2. Total corruption can be injected at each episode is limited |
|  | 3. Number of steps the attacker can inject corruption is limited |
| Goal | Minimize the performance of the policy learned by the agent |

## C    EXPERIMENTAL VALIDATION OF IMPLICATIONS

**Experimental validation of implications** $1$, $2$, $4$, **and 6** We empirically examine the best value of $V$ learned by the agent under our attack methods parameterized with different values of $\Delta$. Note that the value of $|\Delta|$ is also the requirement on $B$ for the attack. To remove the dependency of $V$ on $C$, we set $C = T$, $E = \infty$ so that the attacker is never out of budget due to $C$ and $E$. We choose

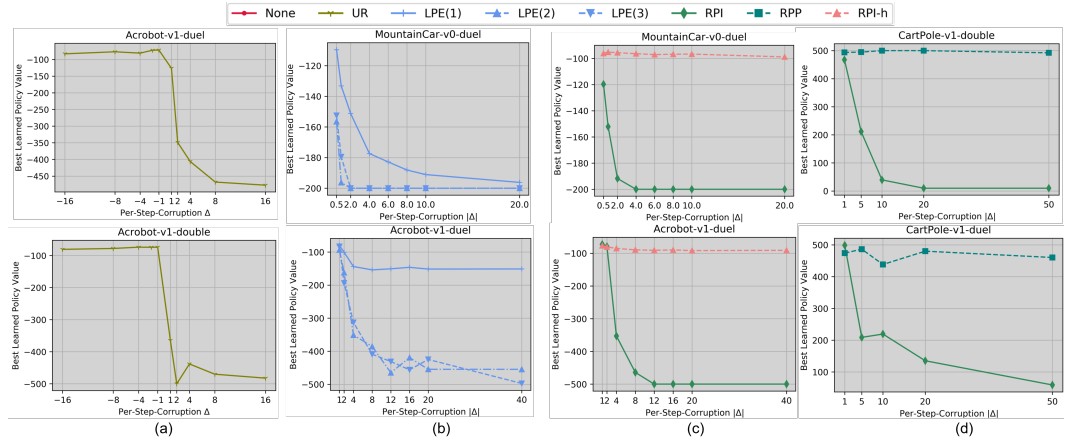

Figure 2: Experimental observations validating implications 1,2,4, and 6 from section 5.

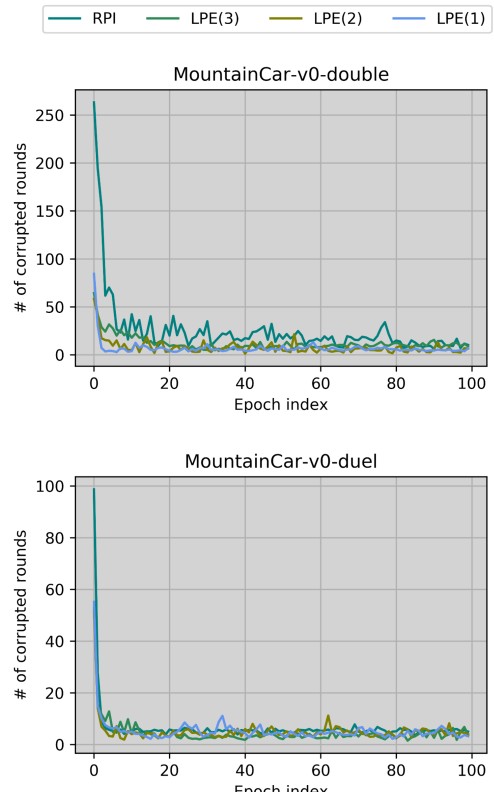

Figure 3: Experimental observations validating implications 3 and 5 from section 5

representative environments to validate our implications in section 5, and the results are shown in figure 2. The x axis is the value of $\Delta$ in (a) the $|\Delta|$ for the rest used by the attacks. The y axis has the same meaning as in figure 1. In figure 2(a) we run the UR attack with both signs of $\Delta$ on environment Acrobot. We observe that when corrupting in the wrong direction $\Delta < 0$, the Duel algorithm can still learn the policies of optimal performance. When corrupting in the right direction $\Delta > 0$, the Duel algorithm learns much worse policies as $\Delta$ increases. This observation agrees with implication 1. In figure 2(b) we run the three variants of the LPE attack on environment MountainCar and Acrobot. We observe that for the same value of $\Delta$, LPE (2) (3) attack can always result in lower policy value than LPE (1) attack, which agrees with implication 2. In figure 2(c) we run the RPI

attack together with its special variant (RPI-h) which uses high performing $\pi^\dagger$. We observe that the RPI attack with a random policy $\pi^\dagger$ always leads to a lower value of the best learned policy compared to the RPI-h with high performing $\pi^\dagger$, which agrees with implication 4. In figure 2(d) we run the RPI and RPP attack with random $\pi^\dagger$ on CartPole. CartPole is an environment that satisfy the condition in implication 6 where the a policy's value is the same as its episode length. We observe that RPI attack leads to very low value of the best learned policy while the RPP attack does not influence the learned policy value no matter how large $|\Delta|$ it uses, which agrees with implication 6.

**Experimental validation of implications 3 and 5**. We experimentally examine the requirement on attack budget $C$ for our attack methods with sufficient value of $|\Delta|$. For a fixed value of $\Delta$, we remove the constraints on $C$ and $E$ so that the corruption can always be applied following $A^t$ defined by different attack methods, and then we measure how many steps are corrupted in each epoch under the attack. The environment we choose is MountainCar, and the value of $|\Delta|$ for both LPE (all the three variants as described in section 6) and RPI attack is 10. The results are shown in figure 3. The x axis is the index of epochs during training, and the y axis is the number of time steps that are corrupted in the epoch. For both LPE and RPI attack, we observe that the number of corrupted steps decrease with time and eventually approaches 0. This suggests that for both attacks, the agent gradually never take actions $a^t$ at the states $s^t$ that correspond to the ones where no currption will be applied under the attack, i.e., $A^t(s^t, a^t) \neq 0$. This observation agrees with implication 3 and 5.

## D    EXPERIMENTS DETAILS AND ADDITIONAL EXPERIMENTS

The hyper parameters for the learning algorithms can be found in the codes. The parameters for the setup of the experiments are given in Table 2. Here the parameter $r$ is the additional parameter for attack against environment with continuous action space as discussed in section 5. The choice on $r$ for the LPE attack and RPI/RPP attack are different. In practice we choose the parameters that significantly reduce the performance of the best learned policy by the learning algorithms. In Table 3 We provide the value of $V_{\min}$, $V_{\max}$, and $L_{\max}$ for each environment we use to determine the constraints on the attack. These value are given by either the setup of the environment or empirically estimation. For example, in MountainCar-v0, $L_{\max} = 200$ and $V_{\min} = -200$ are given by the set up directly, as an episode will be terminated after 200 steps, and the reward is $-1$ for each step. $V_{\max} = -100$ is empirically estimated by the highest reward given by the best policy learned by the most efficient learning algorithm. In Table 4 we provide the policy values (expected total reward from an episode) of $\pi^\dagger$ used by LPE attack (2) and (3). Recall that for LPE attack (2), $\pi^\dagger$ represents an expert policy that have very high performance, and for LPE attack (3), $\pi^\dagger$ represents a median expert policy that also have high performance but less than that for LPE attack (2). The whole set of the main results for our attack methods against learning algorithms under full constraints are shown in figure 4.

Table 2: Parameters for experiments

| ENVIRONMENT | $T$ | $B$ | $E$ | $r$(LPE) | $r$(RPI/RPP) |
|---|---|---|---|---|---|
| CARTPOLE | 80000 | 5 | 500 | / | / |
| LUNARLANDER | 120000 | 4 | 800 | / | / |
| MOUNTAINCAR | 80000 | 2.5 | 200 | / | / |
| ACROBOT | 80000 | 4 | 500 | / | / |
| HALFCHEETAH | 600000 | 42 | 6300 | 2 | 1.5 |
| HOPPER | 600000 | 25 | 2500 | 2 | 1.1 |
| WALKER2D | 600000 | 25 | 2500 | 2.2 | 1.5 |
| SWIMMER | 600000 | 0.8 | 80 | 2.2 | 1.0 |

To make the figures clean, the variance of the results are not included in the figures. We present a subset of variance for our main results as an example in the table 5 below to show that our results are statistically stable.

By the definition 5.8, higher value of $r$ results in higher similarity between policies. By theorem 5.3, the LPE attack should have lower value of $\Delta(V)$ given higher value of $r$; by theorem 5.5 and

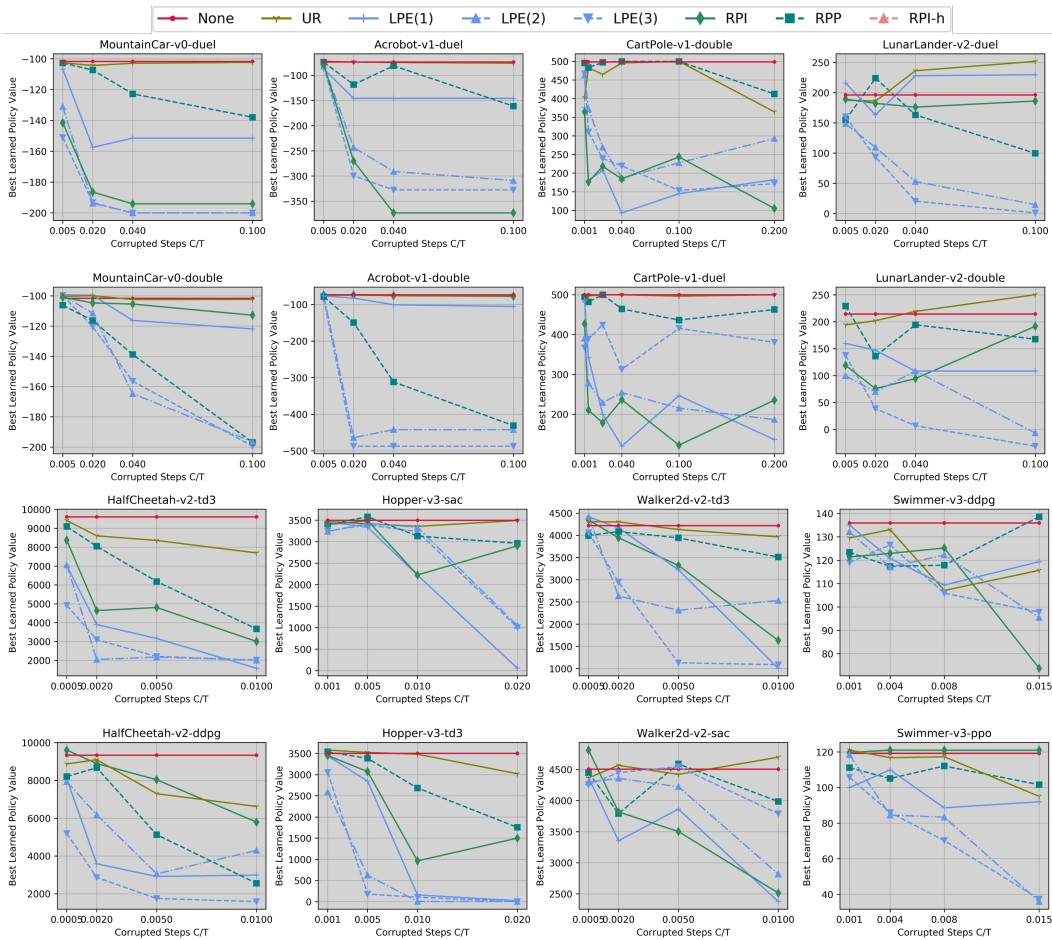

Figure 4: The complete main results for our attack methods against different learning algorithms in different environment

Table 3: Values used to determine the constraints on attack

| ENVIRONMENT | $L_{\max}$ | $V_{\max}$ | $V_{\min}$ | $V_{\max} - V_{\min}$ | $\frac{V_{\max} - V_{\min}}{L_{\max}}$ |
|---|---|---|---|---|---|
| CARTPOLE | 500 | 500 | 0 | 500 | 1 |
| LUNARLANDER | 1000 | 200 | -1000 | 1200 | 1.2 |
| MOUNTAINCAR | 200 | -100 | -200 | 100 | 0.5 |
| ACROBOT | 500 | -100 | -500 | 400 | 0.8 |
| HALFCHEETAH | 1000 | 12000 | 0 | 12000 | 12 |
| HOPPER | 1000 | 4000 | 0 | 4000 | 4 |
| WALKER2D | 1000 | 5000 | 0 | 5000 | 5 |
| SWIMMER | 1000 | 120 | 0 | 120 | 0.12 |

Table 4: Policy value of $\pi^\dagger$ used by LPE attack (2) and (3). Here ALG1 and ALG2 are the pair of learning algorithms we use in each environment. $\mathcal{V}_{\mathcal{M}}^{\pi^\dagger}$-(2)-1 is the policy value of $\pi^\dagger$ for LPE attack (2) when the learning algorithm for the agent is ALG1 (implying that the learning algorithm used by the attacker to learn $\pi^\dagger$ is ALG2). The meanings for the last three columns are similar.

| ENVIRONMENT | ALG1 | ALG2 | $\mathcal{V}_{\mathcal{M}}^{\pi^\dagger}$-(2)-1 | $\mathcal{V}_{\mathcal{M}}^{\pi^\dagger}$-(3)-1 | $\mathcal{V}_{\mathcal{M}}^{\pi^\dagger}$-(2)-2 | $\mathcal{V}_{\mathcal{M}}^{\pi^\dagger}$-(3)-2 |
|---|---|---|---|---|---|---|
| CARTPOLE | DUEL | DOUBLE | 500 | 220 | 500 | 199 |
| LUNARLANDER | DUEL | DOUBLE | 154 | 2 | 202 | 10 |
| MOUNTAINCAR | DUEL | DOUBLE | -108 | -158 | -101 | -156 |
| ACROBOT | DUEL | DOUBLE | -101 | -200 | -100 | -199 |
| HALFCHEETAH | DDPG | SAC | 12374 | 6007 | 12766 | 5974 |
| HOPPER | TD3 | SAC | 3619 | 1828 | 3562 | 1801 |
| WALKER2D | TD3 | SAC | 5172 | 2552 | 4622 | 2426 |
| SWIMMER | DDPG | PPO | 120 | 61 | 120 | 61 |

Table 5: The values in the table are the variance of the performance of the best learned policy over 10 runs in identical settings. The values in the brackets are the corresponding average value that is reported in the main result figure 1.

| Environment-Learning algorithm-Attack | C = 0.001 | C = 0.005 | C = 0.01 | C = 0.02 |
|---|---|---|---|---|
| Hopper-sac-UR | 106(3468) | 114 (3414) | 296(3355) | 90(3492) |
| Hopper-sac-LPE(1) | 125 (3463) | 165 (3335) | 1134 (2220) | 113 (55) |
| Hopper-sac-LPE(2) | 684 (3236) | 131 (3424) | 224 (3332) | 34 (1051) |
| Hopper-sac-LPE(3) | 436(3246) | 860 (3392) | 206 (3231) | 690 (1006) |
| Hopper-sac-RPI | 107(3397) | 81(3489) | 1118(2224) | 628(2901) |
| Hopper-sac-RPP | 132(3407) | 107(3579) | 666(3129) | 604(2961) |
| Hopper-td3-UR | 147(3580) | 212(3524) | 215(3482) | 1032(3021) |
| Hopper-td3-LPE(1) | 133(3443) | 895(2845) | 286(159) | 49(23) |
| Hopper-td3-LPE(2) | 1299(2589) | 1008(626) | 6(5) | 1(4) |
| Hopper-td3-LPE(3) | 1034(3054) | 453(179) | 306(107) | 1(3) |
| Hopper-td3-RPI | 87(3452) | 860(3072) | 206(963) | 690(1501) |
| Hopper-td3-RPP | 175(3538) | 386(3385) | 754(2684) | 870(1755) |

5.7, the RPI and RPP attack should have lower value of $\Delta(V)$ given lower value of $r$. This gives the follow implications:

**Implication** 8**:** Given unlimited buget on $C$ and $E$, with the same value of $|\Delta|$, the LPE attack can make the learning algorithm learn worse policy with higher value of $r$, and the opposite is true for the RPI and RPP attack.

To experimentally validate this implication, we run experiments on Hopper environment and TD3 learning algorithm as an example. We set $C = T$, $E = \infty$, and $|\Delta| = 25$. The results are shown in figure 5. The observation validates implication 8.

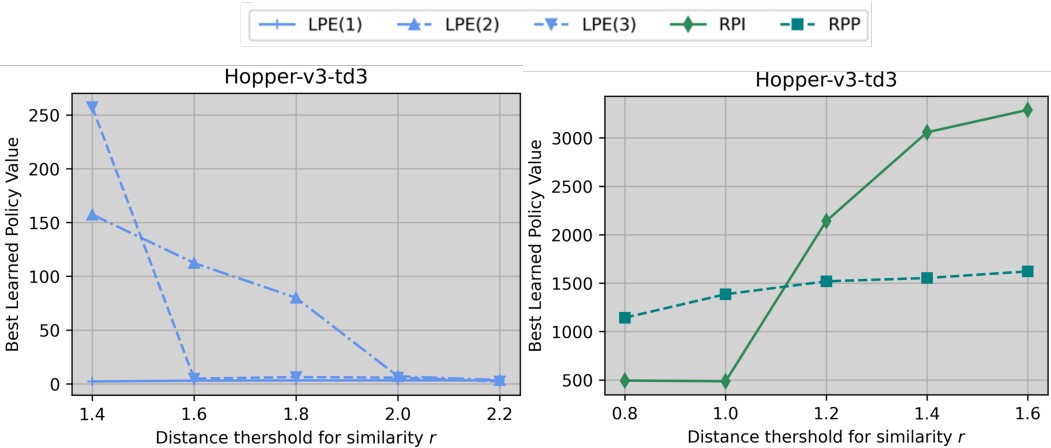

Figure 5: Influence of the distance threshold for similarity $r$ on different attack methods

# E PROOF FOR THEOREMS AND LEMMAS

At the beginning we introduce a simple yet useful lemma for the purpose of simplifying the proof of our theorems and lemmas:

**Lemma E.1.** *A necessary and sufficient condition for the optimal policy $\hat{\pi}^*$ under $\widehat{\mathcal{M}}$ has a policy value less than $V$ under $\mathcal{M}$ is:*

$$\mathcal{V}_{\mathcal{M}}^{\hat{\pi}^*} \leq V \iff \max_{\pi \in B_V^{\mathcal{M}}} \mathcal{V}_{\widehat{\mathcal{M}}}^{\pi} > \max_{\pi \in G_V^{\mathcal{M}}} \mathcal{V}_{\widehat{\mathcal{M}}}^{\pi}.$$

**Proof:** Necessity: When the l.h.s $\mathcal{V}_{\mathcal{M}}^{\hat{\pi}^*} \leq V$ is true, by definition of $B_V^{\mathcal{M}}$ we have $\hat{\pi}^* \in B_V^{\mathcal{M}}$. Since $\widehat{\mathcal{M}}$ is the policy of highest policy value under $\widehat{\mathcal{M}}$, we have $\max_{\pi \in B_V^{\mathcal{M}}} \mathcal{V}_{\widehat{\mathcal{M}}}^{\pi} = \max_{\pi} \mathcal{V}_{\widehat{\mathcal{M}}}^{\pi} > \max_{\pi \in G_V^{\mathcal{M}}} \mathcal{V}_{\widehat{\mathcal{M}}}^{\pi}$. Sufficiency: When the r.h.s $\max_{\pi \in B_V^{\mathcal{M}}} \mathcal{V}_{\widehat{\mathcal{M}}}^{\pi} > \max_{\pi \in G_V^{\mathcal{M}}} \mathcal{V}_{\widehat{\mathcal{M}}}^{\pi}$ is true, then $\hat{\pi}^* \in B_V^{\mathcal{M}}$, and by the definition of $B_V^{\mathcal{M}}$, we have $\mathcal{V}_{\mathcal{M}}^{\hat{\pi}^*} \leq V$.

For convenience, we introduce a definition called value loss to measure the difference between the policy value of a policy under the original and adversarial MDP, :

**Definition E.2.** (Value loss) The value loss of a policy $\pi$ given the environment $\mathcal{M}$ and an adversarial MDP $\widehat{\mathcal{M}}$ is defined as

$$\delta\mathcal{V}_{\mathcal{M},\widehat{\mathcal{M}}}^{\pi} := \sum_{s} \mu^{\pi}(s)(\mathcal{R}(s,\pi(s))) - \hat{\mathcal{R}}(s,\pi(s))$$

, where $\mu^{\pi}(s)$ is the state distribution for the policy $\pi$ representing in expectation how often a state $s$ will be visited in an episode.

The definition of value loss can help us rewriting lemma E.1 as

$$\mathcal{V}_{\mathcal{M}}^{\hat{\pi}^*} \leq V \iff \max_{\pi \in B_V^{\mathcal{M}}} \mathcal{V}_{\mathcal{M}}^{\pi} - \delta\mathcal{V}_{\mathcal{M},\widehat{\mathcal{M}}}^{\pi} > \max_{\pi \in G_V^{\mathcal{M}}} \mathcal{V}_{\widehat{\mathcal{M}}}^{\pi} - \delta\mathcal{V}_{\mathcal{M},\widehat{\mathcal{M}}}^{\pi}. \tag{3}$$

We will frequently using this result for the proof next.

**Proof for theorem 5.1** Under the UR attack, the value loss of a policy is

$$\delta\mathcal{V}_{\mathcal{M},\widehat{\mathcal{M}}}^{\pi} = -p \cdot \Delta \cdot \sum_{s} \mu^{\pi}(s) = -p \cdot \Delta \cdot L^{\pi}.$$

To make the r.h.s in Equation 3 hold, the following needs to be satisfied:

$$\max_{\pi \in B_V^{\mathcal{M}}} V_{\mathcal{M}}^{\pi} + p \cdot \Delta \cdot L^{\pi} > \max_{\pi \in B_V^{\mathcal{M}}} V_{\mathcal{M}}^{\pi} + p \cdot \Delta \cdot L^{\pi}.$$

Equivalently, it requires that there exists $\pi_1 \in B_V^{\mathcal{M}}$ such that for all $\pi_2 \in G_V^{\mathcal{M}}$, the following holds $(V_{\mathcal{M}}^{\pi_1} + p \cdot \Delta \cdot L^{\pi_1}) - (V_{\mathcal{M}}^{\pi_2} + p \cdot \Delta \cdot L^{\pi_2}) > 0$. This can further be formalized as $\min_{\pi_2 \in G_V^{\mathcal{M}}} \max_{\pi_1 \in B_V^{\mathcal{M}}} (V_{\mathcal{M}}^{\pi_1} + p \cdot \Delta \cdot L^{\pi_1}) - (V_{\mathcal{M}}^{\pi_2} + p \cdot \Delta \cdot L^{\pi_2}) > 0$, which gives

$$|\Delta| > \min_{\pi_1 \in B_V^{\mathcal{M}}} \max_{\pi_2 \in G_V^{\mathcal{M}}} \frac{\mathcal{V}_{\mathcal{M}}^{\pi_2} - \mathcal{V}_{\mathcal{M}}^{\pi_1}}{p \cdot |L^{\pi_1} - L^{\pi_2}|}.$$

Then by the definition of $\Delta(V)$, we have $\Delta(V) = \min_{\pi_1 \in B_V^{\mathcal{M}}} \max_{\pi_2 \in G_V^{\mathcal{M}}} \frac{\mathcal{V}_{\mathcal{M}}^{\pi_2} - \mathcal{V}_{\mathcal{M}}^{\pi_1}}{p \cdot |L^{\pi_1} - L^{\pi_2}|}$, and an upper bound can be directly derived as

$$\min_{\pi_1 \in B_V^{\mathcal{M}}} \max_{\pi_2 \in G_V^{\mathcal{M}}} \frac{\mathcal{V}_{\mathcal{M}}^{\pi_2} - \mathcal{V}_{\mathcal{M}}^{\pi_1}}{p \cdot |L^{\pi_1} - L^{\pi_2}|} \leq \min_{\pi_1 \in B_V^{\mathcal{M}}} \max_{\pi_2 \in G_V^{\mathcal{M}}} \frac{V_{\max} - V_{\min}}{p \cdot |L^{\pi_1} - L^{\pi_2}|} = \frac{V_{\max} - V_{\min}}{p \cdot \max_{\pi_1 \in B_V^{\mathcal{M}}} \min_{\pi_2 \in G_V^{\mathcal{M}}} |L^{\pi_1} - L^{\pi_2}|}.$$

**Proof for theorem 5.3**

The value loss for a policy $\pi$ under the LPE attack with policy $\pi^\dagger$ is:

$$\begin{aligned} \delta\mathcal{V}_{\mathcal{M},\widehat{\mathcal{M}}}^{\pi} &= -\sum_s \mu^\pi(s) \cdot \Delta \cdot \mathbb{1}[\pi(s) = \pi^\dagger(s)] \\ &= -\Delta \cdot D(\pi, \pi^\dagger). \end{aligned} \tag{4}$$

Equation 4 says that the value loss for policy $\pi$ is proportional to its similarity with $\pi^\dagger$. To make the r.h.s in Equation 3 hold, the following should be satisfied:

$$\max_{\pi_1 \in B_V^{\mathcal{M}}} V_{\mathcal{M}}^{\pi} + \Delta \cdot D(\pi_1, \pi^\dagger) > \max_{\pi_2 \in G_V^{\mathcal{M}}} V_{\mathcal{M}}^{\pi} + \Delta \cdot D(\pi_2, \pi^\dagger).$$

By similar analysis from proof for theorem 5.1, it can equivalently be rewritten as

$$|\Delta| > \min_{\pi_1 \in B_V^{\mathcal{M}}} \max_{\pi_2 \in G_V^{\mathcal{M}}} \frac{\mathcal{V}_{\mathcal{M}}^{\pi_2} - \mathcal{V}_{\mathcal{M}}^{\pi_1}}{D(\pi_2, \pi^\dagger) - D(\pi_1, \pi^\dagger)} = \Delta(V).$$

Next we give an upper bound on $\Delta(V)$. Let $\pi_0 := \arg\min_{\pi \in B_V^{\mathcal{M}}} D(\pi, \pi^\dagger)$ then the maximum policy value of a policy from $B_V^{\mathcal{M}}$ is lower bound by

$$\max_{\pi \in B_V^{\mathcal{M}}} \mathcal{V}_{\widehat{\mathcal{M}}}^{\pi} = \max_{\pi \in B_V^{\mathcal{M}}} (\mathcal{V}_{\mathcal{M}}^{\pi} - \Delta \cdot D(\pi, \pi^\dagger)) \geq \mathcal{V}_{\mathcal{M}}^{\pi_0} - \Delta \cdot D(\pi_0, \pi^\dagger) \geq V_{\min} - \min_{\pi \in B_V^{\mathcal{M}}} D(\pi, \pi^\dagger)$$

For the maximum policy value of a policy from $G_V^{\mathcal{M}}$, it can be directly upper bound by

$$\max_{\pi \in G_V^{\mathcal{M}}} \mathcal{V}_{\widehat{\mathcal{M}}}^{\pi} = \max_{\pi \in G_V^{\mathcal{M}}} (\mathcal{V}_{\mathcal{M}}^{\pi} - \Delta \cdot D(\pi, \pi^\dagger)) \leq \max_{\pi \in B_V^{\mathcal{M}}} \mathcal{V}_{\mathcal{M}}^{\pi} - \min_{\pi \in B_V^{\mathcal{M}}} \Delta \cdot D(\pi, \pi^\dagger) = V_{\max} - \Delta \cdot \min_{\pi \in B_V^{\mathcal{M}}} D(\pi, \pi^\dagger).$$

Then the upper bound on $\Delta(V)$ can be given as

$$\Delta(V) \leq \frac{V_{\max} - V_{\min}}{\min_{\pi \in G_V^{\mathcal{M}}} D(\pi, \pi^\dagger) - \min_{\pi \in B_V^{\mathcal{M}}} D(\pi, \pi^\dagger)}$$

Note that one can always find a policy $\pi$ that share no similarity to $\pi^\dagger$ by always choosing a different action to $\pi^\dagger$, that is $\pi(s) \neq \pi^\dagger(s), \forall s$, then $D(\pi, \pi^\dagger) = 0$. In the situation where the number of actions is large than 2, then such policy can still have random behavior which usually corresponds to low policy value, and we can assume that a policy with no similarity to $\pi^\dagger$ can always be found in $B_V^{\mathcal{M}}$, suggesting that $min_{\pi \in B_V^{\mathcal{M}}} D(\pi, \pi^\dagger)$, then the upper bound on $\Delta(V)$ can be rewritten as $\Delta(V) \leq \frac{V_{\max} - V_{\min}}{\min_{\pi \in G_V^{\mathcal{M}}} D(\pi, \pi^\dagger)}$.

**Proof for theorem 5.5**

The value loss for a policy $\pi$ under the RPI attack with policy $\pi^\dagger$ is:

$$
\begin{aligned}
\delta \mathcal{V}^\pi_{\mathcal{M}, \widehat{\mathcal{M}}} &= -\sum_s \mu^\pi(s) \cdot \Delta \cdot \mathbb{1}[\pi(s) \neq \pi^\dagger(s)] \\
&= -\sum_s \mu^\pi(s) \cdot \Delta \cdot (1 - \mathbb{1}[\pi(s) = \pi^\dagger(s)]) \\
&= -\sum_s \mu^\pi(s) \cdot \Delta - \sum_s \mu^\pi(s) \cdot \Delta \cdot \mathbb{1}[\pi(s) \neq \pi^\dagger(s)]) \\
&= -\Delta \cdot (L^\pi - D(\pi, \pi^\dagger))
\end{aligned}
\tag{5}
$$

Note that for the attack policy $\pi^\dagger$ itself, its value loss is 0 as $L^{\pi^\dagger} = D(\pi^\dagger, \pi^\dagger)$. To make the r.h.s in Equation 3 hold, the following needs to be satisfied

$$
\max_{\pi_1 \in B_V^{\mathcal{M}}} V^\pi_{\mathcal{M}} + \Delta \cdot (L^{\pi_1} - D(\pi_1, \pi^\dagger)) > \max_{\pi_2 \in G_V^{\mathcal{M}}} V^\pi_{\mathcal{M}} + \Delta \cdot (L^{\pi_2} - D(\pi_2, \pi^\dagger)).
$$

It can be equivalently rewritten as

$$
|\Delta| > \min_{\pi_1 \in B_V^{\mathcal{M}}} \max_{\pi_2 \in G_V^{\mathcal{M}}} \frac{\mathcal{V}^{\pi_2}_{\mathcal{M}} - \mathcal{V}^{\pi_1}_{\mathcal{M}}}{(L^{\pi_2} - L^{\pi_1}) - (D(\pi_2, \pi^\dagger) - D(\pi_1, \pi^\dagger))} = \Delta(V).
$$

Next we give an upper bound on $\Delta(V)$. Since $\pi^\dagger$ is randomly generated, we can assume that it has random behavior in the environment resulting in poor performance, then we can lower bound the the maximum policy value of a policy from $B_V^{\mathcal{M}}$ by

$$
\max_{\pi \in B_V^{\mathcal{M}}} \mathcal{V}^\pi_{\widehat{\mathcal{M}}} = \max_{\pi \in B_V^{\mathcal{M}}} (\mathcal{V}^\pi_{\mathcal{M}} - \Delta \cdot D(\pi, \pi^\dagger)) \geq \mathcal{V}^{\pi^\dagger}_{\mathcal{M}} - \Delta \cdot (L^{\pi^\dagger} - D(\pi^\dagger, \pi^\dagger)) = \mathcal{V}^{\pi^\dagger}_{\mathcal{M}} \geq V_{\min}.
$$

For the maximum policy value of a policy from $G_V^{\mathcal{M}}$, it can be directly upper bound as

$$
\begin{aligned}
\max_{\pi \in G_V^{\mathcal{M}}} \mathcal{V}^\pi_{\widehat{\mathcal{M}}} &= \max_{\pi \in G_V^{\mathcal{M}}} (\mathcal{V}^\pi_{\mathcal{M}} - \Delta \cdot (L^\pi - D(\pi, \pi^\dagger))) \\
&\leq \max_{\pi \in B_V^{\mathcal{M}}} \mathcal{V}^\pi_{\mathcal{M}} - \min_{\pi \in B_V^{\mathcal{M}}} \Delta \cdot (L^\pi - D(\pi, \pi^\dagger)) \\
&= V_{\max} - \Delta \cdot \min_{\pi \in B_V^{\mathcal{M}}} (L^\pi - D(\pi, \pi^\dagger)).
\end{aligned}
$$

Combing both, an upper bound on $\Delta(V)$ can be given as

$$
\Delta(V) \leq \frac{V_{\max} - V_{\min}}{\min_{\pi \in G_V^{\mathcal{M}}} (L^\pi - D(\pi, \pi^\dagger))}.
$$

**Proof for theorem 5.7** Note that the attack strategy for RPP attack has the same form as that of the LPE attack, except that the attack applies positive corruption instead of negative. We can directly write down the policy loss for a policy and $\Delta(V)$ under the RPP attack since they share the same form as those for the LPE attack.

$$
\delta \mathcal{V}^\pi_{\mathcal{M}, \widehat{\mathcal{M}}} = -\Delta \cdot D(\pi, \pi^\dagger).
$$

$$
|\Delta| > \min_{\pi_1 \in B_V^{\mathcal{M}}} \max_{\pi_2 \in G_V^{\mathcal{M}}} \frac{\mathcal{V}^{\pi_2}_{\mathcal{M}} - \mathcal{V}^{\pi_1}_{\mathcal{M}}}{D(\pi_2, \pi^\dagger) - D(\pi_1, \pi^\dagger)} = \Delta(V).
$$

For the upper bound on $\Delta(V)$, we can lower bound the maximum policy value in $\widehat{\mathcal{M}}$ of a policy in $B_V^{\mathcal{M}}$ by that of $\pi^\dagger$, that is,

$$
\max_{\pi \in B_V^{\mathcal{M}}} \mathcal{V}^\pi_{\widehat{\mathcal{M}}} \geq \mathcal{V}^{\pi^\dagger}_{\widehat{\mathcal{M}}} = \mathcal{V}^{\pi^\dagger}_{\mathcal{M}} + \Delta \cdot D(\pi^\dagger, \pi^\dagger) \geq V_{\min} + \Delta \cdot L^{\pi^\dagger}.
$$

The maximum policy value in $\widehat{\mathcal{M}}$ of a policy in $G_V^{\mathcal{M}}$ can be upper bound by

$$\max_{\pi \in G_V^{\mathcal{M}}} \mathcal{V}_{\widehat{\mathcal{M}}}^{\pi} \leq V_{\max} + \Delta \cdot \min_{\pi \in G_V^{\mathcal{M}}} D(\pi, \pi^{\dagger}).$$

Combing both, an upper bound on $\Delta(V)$ can be given as

$$\Delta(V) \leq \frac{V_{\max} - V_{\min}}{L^{\pi^{\dagger}} - \max_{\pi \in G_V^{\mathcal{M}}} D(\pi, \pi^{\dagger})}.$$

**Proof for lemma 5.4** Under the LPE attack, the value loss for any policy is $\geq 0$ as the corruption $\Delta$ is always negative. If a policy $\pi$ has no similarity to the attack policy $\pi^{\dagger}$, that is, $D(\pi, \pi^{\dagger}) = 0$, then its policy loss is $0$. Let $D_0$ be the set of policies that has no similarity to $\pi^{\dagger}$. Then as the value of $|\Delta|$ increases, the value loss for all policies not in $D_0$ increase, then eventually the policy with the highest policy value in $D_0$ will have the highest policy value under the LPE attack with sufficient $\Delta$.

**Proof for lemma 5.6** Under the RPI attack, the value loss for any policy is $\geq 0$ as the corruption $\Delta$ is always negative. The only policy that has $0$ value loss is the attack policy $\pi^{\dagger}$ itself. As the value of $|\Delta|$ increases, the value loss for all policies increase except for $\pi^{\dagger}$, then eventually $\pi^{\dagger}$ will have the highest policy value under the RPI attack with sufficient $\Delta$.

## F    PROS AND CONS OF ATTACK METHODS

To make it clear about which attack method is more promising given a scenario, we summarize the strength and weakness of our attack methods.

**LPE attack:** The main strength is that the attack has less requirement on corruption budget. The attack only applies perturbation for a small subset of actions, so during agent's random exploration, it only applies perturbation for a small portion of the steps. As a result, in most experiments we find that LPE attack usually have significant influence on learning when the attack budget is small. The attack can also benefit from having access to a high performing policy if possible, and in experiments we find that LPE-(2) and (3) are usually more efficient than LPE-(1). The weakness of the attack is that it may fail if the high performing policies can have very different behavior, which can happen if the environment is easy and many policies can be thought of as good. CartPole is a good example of such environments, and in the experiments we can also find that LPE attack is less efficient.

**RPI attack:** Compared to LPE attack, its strength is that it still works even if there are high performing policies of distinct behaviors, as it will only make one policy look the best. This is likely to be the reason why RPI attack is more efficient in CartPole environment. As a cost, it will require more corruption budget, as during agent's random exploration, it applies corruption for most of the time. In experiments we can see that in many scenarios, the RPI attack is more efficient when attack budget is large compared to the LPE attack

**RPP attack:** RPP attack is more efficient than the RPI attack if the agent explores sub-optimal actions more often and vice versa. The reason is that the RPP attack perturbs the steps where the agent explores the optimal actions in the adversarial environment, and the RPI attack does the opposite. For example, in Acrobot and MountainCar environment, we observe that when the learning algorithm is double, RPP is more efficient; and the opposite is true when the learning algorithm is duel.

## G    HARDNESS OF FINDING THE OPTIMAL ATTACK ALGORITHM

First we show the space for all possible attack algorithms is exponentially large in $T$. As discussed in section 3, an attack algorithms can be represented by its attack strategies $A^t$ at each round $t$, and the attack strategy is a mapping $A^t : \mathcal{S}^t \times \mathcal{A}^t \times \mathcal{R}^{t-1} \to \mathcal{C}$, where $\mathcal{C}$ is the corruption space for all possible amount of corruption, $\mathcal{S}^t = \underbrace{\mathcal{S} \times \mathcal{S} \dots \times \mathcal{S}}_{t \text{ times}}$, and the meaning of $\mathcal{A}^t$, $\mathcal{R}^{t-1}$, and $\mathcal{C}^t$ is similar. Given the constraints on the budget of the attacker defined in section 3, finding the optimal attack algorithm requires solving the following optimization problem:

$$\min_{A^{t=1,\ldots,T}} \mathcal{V}^{\pi_0}_{\mathcal{M}}, \text{s.t. } \Delta^t = A^t(s^{1:t}, a^{1:t}, r^{1:t-1}), \sum_{t=1}^{T} \mathbb{1}[\Delta^t \neq 0] \leq C, |\Delta^{t=1,\ldots,T}| \leq B. \tag{6}$$

Even in the tabular setting where the sizes of $\mathcal{S}, \mathcal{A}, \nabla, \mathcal{C}$ are all finite, in total there are $O(|S|^{T^2}|A|^{T^2}|R|^{T^2}|C|^T)$ many possible attack algorithms. This makes exhaustive enumeration computationally infeasible.

Next we show the hardness of searching for the optimal or near optimal attack strategy. Zhang et al. (2020b) show that the poisoning attack problem in the simpler tabular setting can be formulated as an RL problem which is harder than the RL problem for the learning agent. More specifically, the input state of the RL problem for the attacker needs to include the parameters of the learning algorithm, and correspondingly the transition function $\mathcal{P}$ needs to include how such parameters are updated. In the DRL setting, the learning algorithms are more complex compared to the tabular setting. For example, if the learning algorithm is a deep Q learning algorithm, then the input space for the attacker's RL problem need to include all the parameters in the neural networks. Clearly both the input space and transition functions are more complicated in the DRL setting, making the RL problem for the attacker significantly harder to solve regardless of the additional constraints given by attacker's budget.

At last, note that both exhaustive enumeration and the attacker's RL formulation requires that the attacker has full knowledge of both the environment and the learning algorithm which is a strong assumption on attacker's capabilities as mentioned in section 4. Considering all the strong requirements for the attacker and difficulties in finding the optimal or near optimal attack, the goal in our work is to not chase optimality but find feasible attack algorithms that can be constructed efficiently without requiring any knowledge about the environment and the learning algorithm.

## H    EFFICIENCY OF REWARD FLIPPING ATTACK

Here we analyze and empirically examine the performance of a heuristic attack that appears to be effective as believed in Zhang et al. (2021b). The strategy of the attack is to change the sign of the true reward at each time. We call such attack as reward flipping attack, and its attack strategy can be formally written as $A^t(s^t, a^t) = -2r^t$. Note that such attack also construct a stationary adversarial reward function, suggesting that it also falls in our "adversary MDP attack" framework. Under such adversarial MDP $\widehat{\mathcal{M}}$, since all rewards have their signs flipped, we have $\mathcal{V}^{\pi}_{\widehat{\mathcal{M}}} = -\mathcal{V}^{\pi}_{\mathcal{M}}$ for all policy $\pi$, suggesting that the optimal policy under $\widehat{\mathcal{M}}$ actually has the worst performance under the true environment $\mathcal{M}$. However, the disadvantage of the attack method is that it needs to apply corruption at every timestep, resulting in too much requirement on $C$. Note that this is the same issue as the UR attack has. We further empirically test the efficiency of the reward flipping attack. The attacker cannot apply corruption when it runs out of budget on total corruption steps $C$, and we do not assume any constraint on $B$. We consider environment Hopper and HalfCheetah, and set $C = 0.01$ as considered in our experiments while the remaining parameters are unchanged. Our results show that the performance of the reward flipping attack is comparable to the baseline UR attack as shown in the table 6 below, suggesting that the reward flipping attack is not efficient.

Table 6: Performance of reward flipping attack. The values in the table are the performance of the best policy ever learned by the learning algorithm, which is the same as the y axis of our main experiment results in figure 1

| Environment-Learning algorithm | Reward flipping attack | UR attack | No attack |
|---|---|---|---|
| Hopper-td3 | 3157 | 3482 | 3502 |
| Hopper-sac | 3521 | 3355 | 3496 |
| HalfCheetah-ddpg | 7463 | 6622 | 9341 |
| HalfCheetah-td3 | 7603 | 7694 | 9610 |

# I    COMPARISON TO VA2C-P ATTACK

Here we compare the effectiveness of our LPE attack, which is our most efficient attack as an example, and the most "black box" version of VA2C-P attack proposed in Sun et al. (2020) which knows the learning algorithm but does not know the parameters in its model. Note that the constraints for the two attacks are different in the two papers. To make sure that we are not underestimating the effectiveness of the VA2C-P attack, we let both attacks work under the same constraints used in Sun et al. (2020). The constraints here are characterized by two parameters $K$ and $\epsilon$. The training process is separated into $K$ batches of steps, and the attacker is allowed to corrupt no more than $C$ out of $K$ training batches. In each training batch with $t$ time steps, let $\delta \mathbf{r} = \{\delta r_1, \ldots, \delta r_t\}$ be the injected reward corruption at each time step, then the corruption should satisfy $\frac{\|\delta \mathbf{r}\|_2}{\sqrt{t}} \leq \epsilon$. We modify the LPE attack accordingly to work with such constraints. More specifically, the attack strategy is unchanged when applying corruption will not break the constraints, and we forbid the attack to apply corruption if doing so will break the constraints. To avoid anything that could cause a decrease in efficiency of VA2C-P attack, we do not modify any code related to training and attacking provided by Sun et al. (2020), and implement our attack method in their code.

Since VA2C-P has more limitations than LPE attack, we only consider the scenarios where both attack are applicable. As an example, we choose Swimmer as the environment and PPO as the learning algorithm. Here we use the metric in Sun et al. (2020) to measure the performance of the attack. More specifically, we measure the average reward per episode collected by the learning agent through the whole training process. We set $K = 1$, $\epsilon = 1$, and the length of training to be $600$ episodes where each episode consists of $1000$ steps. The results for different attacks are shown in table 7. Each result is the average of 10 repeated experiment, and it is clear that our attack is much more efficient.

Table 7: Comparison between VA2C-P and LPE attack

| clean | VA2C-P | LPE-(1) | LPE-(3) | LPE-(2) |
|-------|--------|---------|---------|---------|
| 30.07 | 25.49  | 14.75   | 7.08    | -2.16   |

We also notice that our LPE attack computes faster than VA2C-P attack. To compute the attack for 600 training episodes in the experiment, our LPE attack takes 135.7 seconds, while the VA2C-P black box attack takes 7049.5 seconds. In this case, the LPE attack computes 52 times faster than VA2C-P attack.

One may notice that learning efficiency of PPO algorithm here is not as good as what we show in Figure 4. This is due to different implementation of the same algorithm in our work and Sun et al. (2020). It is a known issue that difference in implementations can lead to very different learning results Henderson et al. (2018). As mentioned before, we build our learning algorithms based on spinning up documents Achiam (2018), and the learning performance of our learning algorithms match the results shown in the spinning up documents.

