# OpenReview forum: "Efficient Reward Poisoning Attacks on Online Deep Reinforcement Learning"
_ICLR.cc/2023/Conference — Submitted to ICLR 2023_

### Official Review · Reviewer_VFr8 · 2022-10-24

**Confidence:** 4
**Correctness:** 4
**Technical Novelty And Significance:** 3
**Empirical Novelty And Significance:** 3
**Recommendation:** 3

**Clarity, Quality, Novelty And Reproducibility:**

The theoretical results are not presented in a very clear manner, and needs to be improved.

I am not fully convinced that the paper is technically novel enough.

**Strength And Weaknesses:**

Strength:

(1). This paper studied an important topic that lies in the intersection area of reinforcement learning and security, which has important implications in uncovering vulnerability of real-world RL systems and how to design effective defense against attacks.

(2). The paper provided interesting and strong theoretical analysis on the theoretical properties of the proposed attack algorithms.

(3). The paper has solid experiments, and the results clearly demonstrate that the proposed attack algorithms are capable to reducing the performance of the learned RL policy.

Weaknesses:

(1). The presentation of the theoretical results is very confusing.The authors mentioned lower bounds on B, C, and V. However, I am very confused what lower bounds actually mean for equation (1). Does it mean B, C, V has to be at least some number so that equation (1) can be feasible? If so, I think the lower bounds would not make sense because regardless of how small B, C, and V are, one cannot rule out the possibility that there exists some smart and super efficient attacker who can easily corrupt the performance of the learned RL policy with little attack budget. I guess by lower bounds, the authors really mean that B, C, and V, although being as small as some value, the attack can still be successful. I believe the authors really need to rethink how to present their theoretical results in a more clear and succinct manner. Right now it's very confusing and hard to understand.

(2). The attack formulation is not aligned with the attacker ability. The authors mentioned that there is an upper limit E on the per-episode corruption, but this parameter E does not appear in equation (1).

(3). While the authors claimed that the attack does not require knowledge of the underlying environment. In LPE attack, the attacker really needs to first estimate some good policy pi dagger beforehand. This actually requires access to some simulator of the underlying environment. That means, the attack proposed in this paper is not fully black box. The authors need to clarify that in the paper.

(4). My major concern is about the novelty and the significance of contribution. There has been plenty of prior works that studied poisoning attacks on RL, including online RL. The attack of changing the underlying reward function has been investigated in [1] below. Also there was a prior work on attacking online RL [2]. I find it hard to justify how much technical contributions are really significant compared to prior works. Also in the experiment, this paper does not compare their attack algorithms against [2], which is very related work. I think the authors may want to spend more effort discussing how this paper advanced the state-of-the-art attacks against RL and what is really unique about the theory and method of this paper.

Amin Rakhsha, Goran Radanovic, Rati Devidze, Xiaojin Zhu, and Adish Singla. Policy teaching via environment poisoning: Training-time adversarial attacks against reinforcement learning. In International Conference on Machine Learning, pp. 7974–7984. PMLR, 2020.

Xuezhou Zhang, Yuzhe Ma, Adish Singla, and Xiaojin Zhu. Adaptive reward-poisoning attacks against reinforcement learning. In International Conference on Machine Learning, pp. 11225– 11234. PMLR, 2020b.

**Summary Of The Paper:**

This paper studied data poisoning attacks against online reinforcement learning, where an attacker perturbs the reward signal over time to mislead the victim agent into learning a sub-optimal policy. The author provided a generic mathematical formulation of their poisoning attacks (equation (1)), which has three important parameters, including the poisoned policy value V, the total number of poisoning steps C, and the per-step maximum perturbation on the reward signal C. Instead of analyzing the feasibility of attack with respect to V, C, and B, the authors proposed to design attack algorithms and then analyze the corresponding V, C, B that the attack algorithm can induce. In total, the paper proposed four attack algorithms. For each algorithm, the authors analyzed an upper bound on the difference between the perturbed reward function and the original reward function as a function of the desired policy value V. Given this upper bound, the corresponding B can be easily derived. To complement the theoretical analysis, the paper performed empirical study on four deep RL task and several classic RL algorithms, and demonstrated that the proposed attacks can significantly reduce the performance of the best learned RL policy.

**Summary Of The Review:**

I work on related areas.

---

> ### Author Response · Authors · 2022-11-16
> **Response to the questions**
>
> Q1: The authors mentioned lower bounds on B, C, and V. However, I am very confused what lower bounds actually mean for equation (1).
>
> R1: ‘I guess by lower bounds, the authors really mean that B, C, and V, although being as small as some value, the attack can still be successful.’ This part is an accurate understanding. In the main text we mention that ‘Next, we compute bounds on V, B, and C such that $A^{1:T}$ constructed using equation 2 from a given $\hat{\mathcal{M}}$ in our framework is a feasible solution to equation 1’. We want to find a feasible solution through an attack method to equation (1) with parameters B, C, V. It is trivial to show that if a solution (given by an attack method) satisfies a tuple of (B,C,V), then it also satisfies higher values of B, C, V. So generally speaking we want to find the minimal value of B, C, V that a solution can satisfy for a given attack to measure its efficiency, and this is the reason why we study the lower bound.
>
> Q2: The authors mentioned that there is an upper limit E on the per-episode corruption, but this parameter E does not appear in equation (1), In LPE attack, the attacker really needs to first estimate some good policy pi dagger beforehand.
>
> R2: We mention that for theoretical analysis, E will not be discussed for simplicity, otherwise it makes the problem too complicated to analyze. Note that the conclusion of our analysis still holds in experiments, even though we did not model E.
>
> Q3: There has been plenty of prior works[1,2] that studied poisoning attacks on RL, including online RL.
>
> R3: All works mentioned here are about data poisoning attacks in the tabular MDP setting, which are not applicable to the more complicated deep MDP setting considered in our work. This fact is already mentioned in our related works section starting with ‘Data poisoning attack on bandit and tabular RL settings’, where you can find more details.
>
> References:
>
> [1]Amin Rakhsha, Goran Radanovic, Rati Devidze, Xiaojin Zhu, and Adish Singla. Policy teaching via environment poisoning: Training-time adversarial attacks against reinforcement learning. In International Conference on Machine Learning, pp. 7974–7984. PMLR, 2020.
>
> [2] Xuezhou Zhang, Yuzhe Ma, Adish Singla, and Xiaojin Zhu. Adaptive reward-poisoning attacks against reinforcement learning. In International Conference on Machine Learning, pp. 11225– 11234. PMLR, 2020b.

---

### Official Review · Reviewer_de9h · 2022-10-24

**Confidence:** 2
**Correctness:** 4
**Technical Novelty And Significance:** 3
**Empirical Novelty And Significance:** 3
**Recommendation:** 5

**Clarity, Quality, Novelty And Reproducibility:**

- Clarity: there are no problems with clarity, although it's also not an example of great presentation.

- Quality: quality seems fine. In some sense it should be very non-surprising that if an adversary can corrupt reward in an MDP, then they can make the policy learn the wrong thing.

- Novelty: adversarial attacks in RL are not a hugely explored area, and this work seems to take a novel perspective on it.

- Reproducibility: enough information seems to be given to reproduce the results. A quick review of the code in the supplementary material seems reasonable.

**Strength And Weaknesses:**

Strengths: the proposed methods for coming up with attacks are intuitive yet clever, and the general idea and empirical evaluation seem convincing.

Weaknesses: the most obvious weakness is that it seems totally unsurprising that, if an adversary is able to change the rewards in an MDP, they should have a lot of power to make the agent learn the wrong policy. ~~On the one hand, this makes it sort of a less interesting example of an adversarial attack (it's like corrupting labels would be for supervised learning).~~ On the other hand, maybe it's actually more realistic because adding some adversarial noise to the reward signal might be very possible in some circumstances.

**Summary Of The Paper:**

The authors consider poisoning attacks on RL agents (i.e. adversarially modify the training data to harm performance) in a setting where the poisoner can only corrupt the reward, and with an ability to do this that is budget-limited in various ways. The authors frame this problem as the goal of finding a new MDP (by changing the reward function only) whose optimal policy will be the one desired by the attacker. Although this is computationally hard, the authors propose some good heuristics for solving it, and empirically experiment with them.

**Summary Of The Review:**

Overall, the paper seems fine; the ideas in it are argued and presented reasonably but don't seem very surprising or profound. As such, I think it seems like a fine paper but probably not enough to get accepted to ICLR. Because I do not know the area well, my confidence is low and I could be persuaded in either direction.

---

> ### Author Response · Authors · 2022-11-16
> **Response to the questions**
>
> Q1: On the one hand, this makes it sort of a less interesting example of an adversarial attack (it's like corrupting labels would be for supervised learning).
>
> R1: There is a misunderstanding. Reinforcement learning and supervised learning are intrinsically different learning processes, so the attacks influence the learning process in the two settings in fundamentally different manners.
>
> First, reinforcement learning considered in our work is an online learning process in contrast to the offline learning process in supervised learning. In an online learning process, the learner has to collect data in a strategic manner due to the well-known problem of the ‘exploration-exploitation trade-off’. The data poisoning attack needs to consider how the perturbation can influence the exploration of the agent in a reinforcement learning setting, which is not considered in attacks against supervised learning.
>
> Second, in the supervised learning setting, all true labels are given at the beginning. In the reinforcement learning setting, though the goal of the agent is to learn to output an optimal ‘label’ for different inputs, there is no true label but only an instant temporary reward, and the optimization target is the summation of the temporary rewards over a sequence of actions. Therefore, in reinforcement learning, the attack only perturbs very indirect signals for the agent to learn from, while in supervised learning, the attack directly perturbs the target that the agent wants to generalize.

---

> > ### Comment · Reviewer_de9h · 2022-11-24
> > **Thank you for clarification**
> >
> > Thank you for this clarification. While I don't think it ultimately changes the conclusion of that part of my review (which is pretty low confidence in any case), I will strike out that sentence as you have convinced me this is not a good analogy.

---

### Official Review · Reviewer_oGPF · 2022-10-25

**Confidence:** 4
**Correctness:** 3
**Technical Novelty And Significance:** 2
**Empirical Novelty And Significance:** 2
**Recommendation:** 3

**Clarity, Quality, Novelty And Reproducibility:**

Clarity: the paper is clear about its goal but less clear about its thought flow. For example, why the authors propose the four different attacks, and the pros and cons of these methods. The figure 1 is also not very clear in terms of what the authors want to imply and what conclusion should we draw. It's clear from the figure that all the attacks are successful to bring the performance down, and the case without attack has the best policy return. However, the used markers to represent different classes are hard to distinguish (e.g. LPE2 and LPE3), and the points need to have error bars, and the fonts need to be larger to be readable.

Quality: the presented algorithms are working well, but are not discussed thoroughly, both in terms of the algorithms themselves and in terms of the results on these algorithms. The authors did say algorithm 1 is better/worse than algorithm 2, but did not provide convincing explanations.

Novelty: since the claims in the paper are kind of over-claiming their contributions, it is not clear how much novelty the work has.

Reproducibility: code is provided, not sure if the results would match because we don't have the error bars in the figure. Also, the y-axis in figure 1 says policy value, is that the total episodic rewards or really the policy value? Usually the value is associated with a starting state, what is the starting state?

**Strength And Weaknesses:**

The work presents an interesting study on reward poisoning attacks where the attacker does not know the training algorithm of the victim agent. The main novelty is that it does not require knowing the victim agent's policy training details, in particular the training algorithm and related hyper parameters, while other previous works may require this. However, there are several weakness of the work.

First, some descriptions over-claims the contribution of the work. For example, the author claimed the attacker does not know the training algorithm of the victim agent, however, in the experiment section, the work only evaluates the case where the attacker and the victim agent use different training algorithms, without evaluating the case when they use the same algorithm. The author also claims the attacker knows no detailed knowledge of the agent's environment, but can observe the full observation, actions, rewards generated during training at each time step, and these information already leaks more than enough information of the training environment, and the attacker may use this knowledge to perform other kinds of adversarial attack, even when the attacker does not know the training algorithm of the victim agent. Similarly, in the experiment section, the attacker has access to environment to learn attack policy offline, which means the attacker has access to the environment, and this is not consistent with the claim that the attacker does not have detailed knowledge of the environment.

Second, the organization and/or thought flow of the paper is not carefully designed. The author first proposes a very general optimization problem to solve to find the feasible attacks, but then transits to discuss several weakly-related attacks. For example, the uniformly random time attack can serve as a good baseline to attack the victim with randomly proposed reward perturbation, however, it is not clear why the author chooses to talk about learned policy evasion attack as the immediate next attack. For example, the paper can talk about the limitation of UR attack, and then propose LPE attack that addresses part of the limitations. Similarly, why the RPI attack is proposed following LPE attack, and so on.

Third, the experiments are limited and not thoroughly discussed. The paper spends the majority of its pages discussing the theoretical insights of the different methods, however, does not discuss intuitively the pros and cons of the various proposed algorithms. This making it unclear why these methods are proposed and under what circumstances should the attacker use what algorithm. If one algorithm strongly outperforms the other three, why proposing the other algorithms, etc. The data in figure 1 is also not discussed well, and ideally should have error bar.

**Summary Of The Paper:**

The work designed a set of black-box adversarial attacks to corrupt a small proportion of training time rewards, and make the agent learn a low-performing policy. The goal and contribution of this work is on efficient poisoning attacks on DRL via reward poisoning, assuming the attacker has no knowledge of the exact DRL algorithm and does not have detailed knowledge about the agent's environment, and the attacker has limits on the amount of the reward corruption it can apply. The formulation of the problem is to find a training time reward perturbation such that the resultant learned policy has a bad performance, and also under the limit of the times of reward corruption and magnitude of reward corruption. However, realizing that directly solving the optimization problem is computationally infeasible. The work proposes to evaluate the limit of any reward poisoning attack algorithm on its attack performance, limits on reward perturbation times and magnitude. The trade-off between limits on reward perturbation times and magnitude as well as resultant policy value affects the number of available adversarial MDP that satisfies the constraints. The work then proposes several practical adversarial attacks on rewards. Uniformly random time attack as a baseline, where the attacker randomly perturb reward with a fixed probability and fixed amount. Learned policy evasion attack: the attacker penalizes the learner whenever it chooses a good action that are shared between several good policies. Random policy inducing attack: a fixed reward perturbation is added whenever the agent behaves differently from a random policy. Random policy promoting attack: rewards positively when the agent chooses actions the same as a random policy. Experiment study on these attacks for DQN, Double DQN, DDPG, TD3, SAC, PPO are presented to demonstrate the performance of these attacks.

**Summary Of The Review:**

To sum, this work presents a relatively novel set of attack methods on reward poisoning for training time attack on DRL. The contribution of the work is limited and unclear. The experiments require a lot of training and work, but can still be improved and probably should have more space in the paper. Overall, this is an interesting work with a good potential but probably needs more work to improve the writing and experiments.

---

> ### Author Response · Authors · 2022-11-16
> **Response to the questions**
>
> Q1: The organization and/or thought flow of the paper is not carefully designed
>
> R1: We believe that the organization of our paper is proper. We first formulate the problem of finding the optimal attack as an optimization problem and argue that it is infeasible to solve the problem exactly. Then we turn to find a feasible attack and formulate a general attack framework as our basic idea to solve the problem. Next, we present several instances of the attack framework. We start with a naive baseline called the UR attack that applies corruption at random times, and we point out its limitations. Then we construct our main attack methods in a more strategic manner, and we further analyze how efficient these attack methods should be. In the final section, we empirically verify the efficiency of the attack methods we propose.
>
> Q2: The experiments are limited and not thoroughly discussed
>
> R2: Due to the complexity of the deep reinforcement learning problem, our theoretical analysis of our attack methods cannot predict one attack method to be strictly better than another. We have to verify the efficiency of our attack methods in different scenarios empirically. In this case, we propose multiple attack methods as different options, and the empirical results indicate which method is more likely to work well in different scenarios. We have added more discussion about the potential pros and cons of our attack and relate them to the experimental results, which can be found in Appendix F in the latest version.
>
> Q3: code is provided, not sure if the results would match because we don't have the error bars in the figure. Also, the y-axis in figure 1 says policy value, is that the total episodic rewards or really the policy value? Usually the value is associated with a starting state, what is the starting state?
>
> R3: The results of the variance of the statistics were already provided in Appendix D and one can reproduce the results using the same random seed. In our main text, we have already defined the meaning of:
>
> 1. y-axis in figure 1 as: ‘the y axis is the policy value of the best policy the learning algorithm learned after each epoch across the whole training process’.
>
> 2. policy value as: ‘We denote $\mathcal{V}^\pi_{\mathcal{M}} := E_{s_0 \sim \mu}\mathcal{V}^\pi_{\mathcal{M}}(s_0)$  as the policy value for a policy $\pi$ in $M$, which measures the performance of $\pi$.’.
>
> 3. The starting state as ‘μ is the distribution of the initial states’. In addition, note that the distribution of initial states is part of the definition of the Markov decision process that characterizes the environment.

---

### Author Response · Authors · 2022-11-16
**General response to the common concern about our black box setting**

Dear area chair and reviewers,

Thanks for your feedback. We realize that the reviewers misunderstood our black box setting. We have added extra clarifications (in red) to make these clearer in the updated paper. In the general response presented here we provide clarifications to the specific concerns raised by the reviewers in this direction.

Q1 (Reviewer oGPF):  For example, the author claimed the attacker does not know the training algorithm of the victim agent, however, in the experiment section, the work only evaluates the case where the attacker and the victim agent use different training algorithms, without evaluating the case when they use the same algorithm.

R1: All of our attacks except LPE(2) and (3) do not use any learning algorithm. Our attacks other than LPE(2) and (3) are already efficient without having access to a high-performing policy. We study LPE(2) and (3) to show the impact of extra access on attack efficiency.
For LPE(2) and LPE(3), to avoid utilizing any information about the learning algorithm, we rely on a policy learned with a different algorithm to represent the general case that the attacker chooses an arbitrarily efficient learning algorithm that is highly likely to be different from the one used by the learning agent.

Q2 (Reviewer oGPF): The author also claims the attacker knows no detailed knowledge of the agent's environment but can observe the full observation, actions, rewards generated during training at each time step, and this information already leaks more than enough information of the training environment.

R2. We want to clarify it is a misunderstanding that observing the interaction of an agent and an environment is roughly equivalent to having full knowledge of the environment. In fact, the gap between the two knowledge settings is huge.

In the RL setting, an agent’s knowledge of the environment can be broadly classified into four well-known scenarios 1. The world model of the environment [1] 2. A simulator of the environment [2] 3. Offline data set collected in the environment [3] 4. Observation of another agent interacting with the environment [4]. The world model contains full information about the transition and reward function of the MDP corresponding to the environment. If one only has access to a simulator that can sample from the environment, it will take an exponentially large number of samples to approximate the world model even in the finite tabular MDP setting [5]. Suppose one only has access to an offline dataset of observations of the environment. In that case, the difficulty for it to learn from the offline dataset is significantly harder than learning from a simulator in an online manner [3]. If one is only able to observe the interaction as in our case, one has to make decisions at the beginning with a small size of the dataset collected from the observations. Combining all together, the gap between 1 and 4 is huge, and therefore the learning agent in our case does not leak enough information about the environment. We note that we do not learn from this dataset of observations to make attack decisions.

Reference:

[1]: Sutton, Richard S., and Andrew G. Barto. Reinforcement learning: An introduction.

[2]: Todorov, Emanuel, Tom Erez, and Yuval Tassa. "Mujoco: A physics engine for model-based control." 2012 IEEE/RSJ international conference on intelligent robots and systems. IEEE, 2012

[3]: Levine, Sergey, et al. "Offline reinforcement learning: Tutorial, review, and perspectives on open problems." arXiv preprint arXiv:2005.01643 (2020).

[4]:Sun, Yanchao, Da Huo, and Furong Huang. "Vulnerability-aware poisoning mechanism for online rl with unknown dynamics." arXiv preprint arXiv:2009.00774 (2020).

[5]:Agarwal, Alekh, et al. "Reinforcement learning: Theory and algorithms." CS Dept., UW Seattle, Seattle, WA, USA, Tech. Rep (2019): 10-4.

Q3 (Reviewer oGPF): Similarly, in the experiment section, the attacker has access to the environment to learn attack policy offline, which means the attacker has access to the environment, and this is not consistent with the claim that the attacker does not have detailed knowledge of the environment.

(Reviewer vFR8): While the authors claimed that the attack does not require knowledge of the underlying environment. In LPE attack, the attacker really needs to first estimate some good policy pi dagger beforehand.)

R3.  In the background section, we state that our attack methods do not require knowledge of the environment. In the attack methods section, we study extra scenarios (the LPE(2) and (3) attacks) where an attacker could benefit from having access to a high-performing policy.

---

### Decision · Program_Chairs · 2023-01-20

**Decision:**

Reject

**Justification For Why Not Higher Score:**

The reviewers pointed out several weaknesses in the paper; there was a consensus in the reviewers' ratings for rejection.

**Justification For Why Not Lower Score:**

N/A

**Metareview: Summary, Strengths And Weaknesses:**

The reviewers agreed that the paper investigates an interesting setting of reward poisoning attacks where the attacker doesn't know the agent's training algorithm. However, the reviewers pointed out several weaknesses in the paper and shared common concerns. We want to thank the authors for their detailed responses; unfortunately, the final decision is a rejection. The reviewers have provided detailed and constructive feedback. We hope that the authors can incorporate this feedback when preparing future revisions of the paper.